# NFATc1 marks articular cartilage progenitors and negatively determines articular chondrocyte differentiation

**Fan Zhang[1,2†], Yuanyuan Wang[1,2†], Ying Zhao[1], Manqi Wang[3], Bin Zhou[4], Bin Zhou[5], Xianpeng Ge[1,2]\***

[1]Xuanwu Hospital, Capital Medical University, Beijing, China; [2]National Clinical Research Center for Geriatric Diseases, Beijing, China; [3]Central South University, Changsha, China; [4]Department of Genetics, Pediatrics, and Medicine (Cardiology), Albert Einstein College of Medicine of Yeshiva University, New York, United States; [5]Shanghai Institute of Biochemistry and Cell Biology, Chinese Academy of Sciences, Shanghai, China

**\*For correspondence:**
xianpeng.ge@xwhosp.org;
xpge@penggegroup.com

[†]These authors contributed equally to this work

**Competing interest:** The authors declare that no competing interests exist.

**Abstract** The origin and differentiation mechanism of articular chondrocytes remain poorly understood. Broadly, the difference in developmental mechanisms of articular and growth-plate cartilage is still less elucidated. Here, we identified that the nuclear factor of activated T-cells cytoplasmic 1 (NFATc1) is a crucial regulator of articular, but not growth-plate, chondrocyte differentiation during development. At the early stage of mouse knee development (embryonic day 13.5), NFATc1-expressing cells were mainly located in the flanking region of the joint interzone. With development, NFATc1-expressing cells generated almost all articular chondrocytes but not chondrocytes in limb growth-plate primordium. NFATc1-expressing cells displayed prominent capacities for colony formation and multipotent differentiation. Transcriptome analyses revealed a set of characteristic genes in NFATc1-enriched articular cartilage progenitors. Strikingly, the expression of NFATc1 was diminished with articular chondrocyte differentiation, and suppressing NFATc1 expression in articular cartilage progenitors was sufficient to induce spontaneous chondrogenesis while overexpressing NFATc1 suppresses chondrogenesis. Mechanistically, NFATc1 negatively regulated the transcriptional activity of the *Col2a1* gene. Thus, our results reveal that NFATc1 characterizes articular, but not growth-plate, cartilage progenitors during development and negatively determines articular chondrocyte differentiation at least partly through regulating COL2A1 gene transcription.

## Editor's evaluation

NFATc1 was known as a crucial regulator in osteoclast differentiation. The current study presented surprising and novel findings, showing the specific expression of NFATc1 in articular cartilage and its function in cartilage biology. This is a significant discovery since it will help us understand the regulatory mechanism of articular chondrocyte differentiation and the development of osteoarthritis disease.

## Introduction

The basic mechanism underlying articular cartilage development, particularly the origin and differentiation of articular chondrocytes, remains poorly understood. It is well appreciated that synovial joint tissues, including the articular cartilage, originate from a distinct group of progenitors from those that generate the limb primary cartilaginous anlagen *Koyama et al., 2008*; *Chijimatsu and Saito, 2019*.

**eLife digest** Within the body are about 300 joints connecting bones together. Many factors – including trauma, inflammation, aging, and genetic changes – can affect the cushion tissue covering the end of the bones in these joints known as articular cartilage. This can lead to diseases such as osteoarthritis which cause chronic pain, and in some cases disability.

To treat such conditions, it is essential to know how cells in the articular cartilage are formed during development. In the embryo, most cells come from groups of progenitor cells that are programmed to produce specific types of tissue. But which progenitor cells are responsible for producing the main cells in articular cartilage, chondrocytes, and the mechanisms that govern this transformation are poorly understood.

In 2016, a group of researchers found that the gene for the protein NFATc1, which is important for building bone, is also expressed in a group of progenitor cells at the site where ligaments insert into bone in mice. Inactivation of NFATc1 in these progenitor cells has also been shown to cause abnormal cartilage to form, a condition termed osteochondromas. Building on this work, Zhang, Wang et al. – including some of the researchers involved in the 2016 study – set out to find whether NFATc1 is also involved in the normal development of articular chondrocytes.

To investigate, the team used genetically modified mice in which any cells with NFATc1 also had a green fluorescent protein, and tracked these cells and their progeny over the course of joint development. This led them to discover a group of NFATc1-containing progenitor cells that gave rise to almost all articular chondrocytes in the knee joint.

Further experiments revealed that when NFATc1 was removed, this made the progenitors become articular chondrocytes very quickly. In contrast, when the cells had excess amounts of the protein, the formation of articular chondrocytes was significantly reduced. This suggests that the level of NFATc1 governs when progenitors develop into articular chondrocytes.

These findings have provided a way to track the progenitors of articular chondrocytes throughout development and study how articular cartilage is formed. In the future, this work could help researchers develop treatment strategies for osteoarthritis and other cartilage-based diseases. However, before this can happen, further work is needed to confirm that the effects observed in this study also relate to humans.

GDF5 is one of the genes widely used for tracking synovial joint and articular cartilage development *Decker, 2017*. Using reporter mice, multiple groups have displayed that *Gdf5*-expressing cell lineages form almost all articular chondrocytes *Koyama et al., 2008*; *Rountree et al., 2004*; *Shwartz et al., 2016*; *Decker et al., 2017*. As GDF5 is expressed in both interzone cells and its flanking cells at the early stage of joint morphogenesis, current results cannot discriminate which site is the origin of articular chondrocytes. Also, GDF5 expression is greatly diminished at the late stage of embryonic development and almost undetectable in articular cartilage in neonatal mice *Decker, 2017*. Thus, GDF5 cannot be used to track articular cartilage progenitors postnatally.

PRG4 is an articular cartilage progenitor marker in the late stage of synovial joint development *Chijimatsu and Saito, 2019*; *Chagin and Medvedeva, 2017*. This gene encodes lubricin, a major component of synovial fluid and responsible for joint lubricity *Coles et al., 2010*. PRG4 is detected from the stage of joint cavitation and is predominantly expressed in the surficial layer of developed articular cartilage *Rhee et al., 2005*. Several studies exploited *Prg4^CreERT2* reporter mice to track postnatal articular cartilage development and identified this gene as a marker for postnatal and adult articular cartilage progenitors *Decker et al., 2017*; *Kozhemyakina et al., 2015*. Since PRG4 starts to predominantly express and function at the late stage of articular cartilage development, it does not label the primary progenitors of articular cartilage. Several other molecules, such as *Sox9, Dkk3, and Tgfbr2*, were also utilized to track articular cartilage and synovial joint development *Shwartz et al., 2016*; *Decker et al., 2017*; *Li et al., 2013*, but none of these molecules have been shown to specifically and constantly label articular cartilage progenitors and to be able to distinguish the origin of articular chondrocytes.

In addition to the origin of articular chondrocytes, molecular mechanisms regulating articular chondrocyte differentiation remains largely unknown. In particular, the transcriptional regulation of articular

chondrocyte differentiation is far from clear. SOX9 is essential in multiple steps of chondrogenesis, but it was originally and mainly studied in growth-plate chondrocytes *Lefebvre and Dvir-Ginzberg, 2017*. Although SOX9 is also expressed in articular cartilage and is essential for maintaining adult articular cartilage homeostasis *Haseeb et al., 2021*, its detailed functions and mechanisms in articular cartilage development remain to be elucidated. Also, SOX9 starts to express in mesenchymal cells from the very early stage of limb development before the cartilage template formation and it alone is not sufficient to induce chondrogenesis *Lefebvre and Dvir-Ginzberg, 2017*; *Akiyama et al., 2005*. Therefore, the identification of a core transcriptional regulator of articular chondrocyte differentiation is paramount for understanding the basic mechanism of articular cartilage development and exploring new strategies for treating disorders of articular cartilage.

The nuclear factor of activated T-cells, cytoplasmic 1 (NFATc1) is one of the five members of the NFAT family, which share a similar DNA binding domain of approximately 300 amino acid residues *Hogan et al., 2003*; *Vaeth and Feske, 2018*. NFAT signaling plays a broad function in various physiological and pathological processes, including immune cell differentiation and functions, cardiac valve development, and cancer progression and metastasis *Hogan et al., 2003*; *Mancini and Toker, 2009*. In the skeletal system, NFATc1 is critical for osteoclast differentiation and functions *Aliprantis et al., 2008*; *Takayanagi, 2007* and is also involved in osteoblast differentiation by cooperating with the Osterix gene *Koga et al., 2005*. Intriguingly, NFATc1 expression was found in the superficial layers of articular cartilage as well and decreased in human osteoarthritic cartilage *Greenblatt et al., 2013*. Following these studies, we recently identified a function of NFATc1 in restricting osteochondroma formation from entheseal progenitors *Ge et al., 2016*, revealing that NFATc1 is a suppressor of chondrogenesis in these cells.

In this study, we unexpectedly found that NFATc1 constantly labels articular cartilage progenitors throughout embryonic development and postnatal growth. The expression of NFATc1 is diminished with articular chondrocyte differentiation, and suppression of NFATc1 in articular cartilage progenitors is sufficient to induce spontaneous chondrocyte differentiation through regulating the transcriptional activity of the *Col2a1* gene. These findings provide novel insights into the identity and origin of articular cartilage progenitors and identify a fundamental function of NFATc1 in determining physiological articular chondrocyte differentiation.

## Results

### Articular cartilage is derived from NFATc1-expressing progenitors

Following our previous discovery that NFATc1 identifies entheseal progenitors at the site of ligaments inserted onto the bone *Ge et al., 2016*, we unexpectedly found that in $Nfatc1^{Cre}$;$Rosa26^{mTmG/+}$ dual-fluorescence reporter mice, the majority of articular chondrocytes expressed green fluorescence protein (GFP) at 8 weeks of age [*Figure 1(A and B)*, 90.55 ± 6.38%, n=5 mice]. As in this genetic reporter mouse line, both *Nfatc1*-expressing cells and their progenies express GFP, this finding suggests that articular chondrocytes were either expressing *Nfatc1* or derived from *Nfatc1*-expressing progenitors. To clarify the expression pattern of NFATc1 during articular cartilage development, we mapped GFP$^+$ cells in $Nfatc1^{Cre}$;$Rosa26^{mTmG/+}$ mice at the early stage of knee joint morphogenesis (E13.5), postnatal day 0 (P0), and 2 weeks of age [*Figure 1(A)*]. At E13.5, GFP$^+$ cells were mainly localized to the flanking region of the joint interzone with only sporadic distribution in the interzone site. We further examined the expression of NFATc1 at this stage by crossing the tamoxifen-induced $Nfatc1^{CreERT2}$ mouse line with $Rosa26^{mTmG/+}$ mice to generate $Nfatc1^{CreERT2}$;$Rosa26^{mTmG/+}$ reporter mice, in which the real-time expression of NFATc1 could be reflected by GFP shortly after tamoxifen pulse. The localization of *Nfatc1*-expressing cells surrounding the joint interzone was verified after administering tamoxifen to $Nfatc1^{CreERT2}$;$Rosa26^{mTmG/+}$ mice at E11.5 and sampling at E13.5 [*Figure 1—figure supplement 1A*].

In neonatal $Nfatc1^{Cre}$;$Rosa26^{mTmG/+}$ mice (P0), GFP$^+$ cells consisted of a portion of cells in the presumptive articular cartilage site [*Figure 1A*]. Strikingly, at 2 weeks of age, most articular chondrocytes turned out to be GFP$^+$, similar to that at 8 weeks of age [*Figure 1(A)*]. To clarify the real-time expression of NFATc1 in articular cartilage at 2 weeks and 8 weeks of age, we used $Nfatc1^{CreERT2}$;$Rosa26^{RFP/+}$ reporter mice, in which the expression of NFATc1 is reflected by red fluorescence protein (RFP) shortly after tamoxifen administration. Of interest, with tamoxifen pulse, RFP$^+$ cells were found

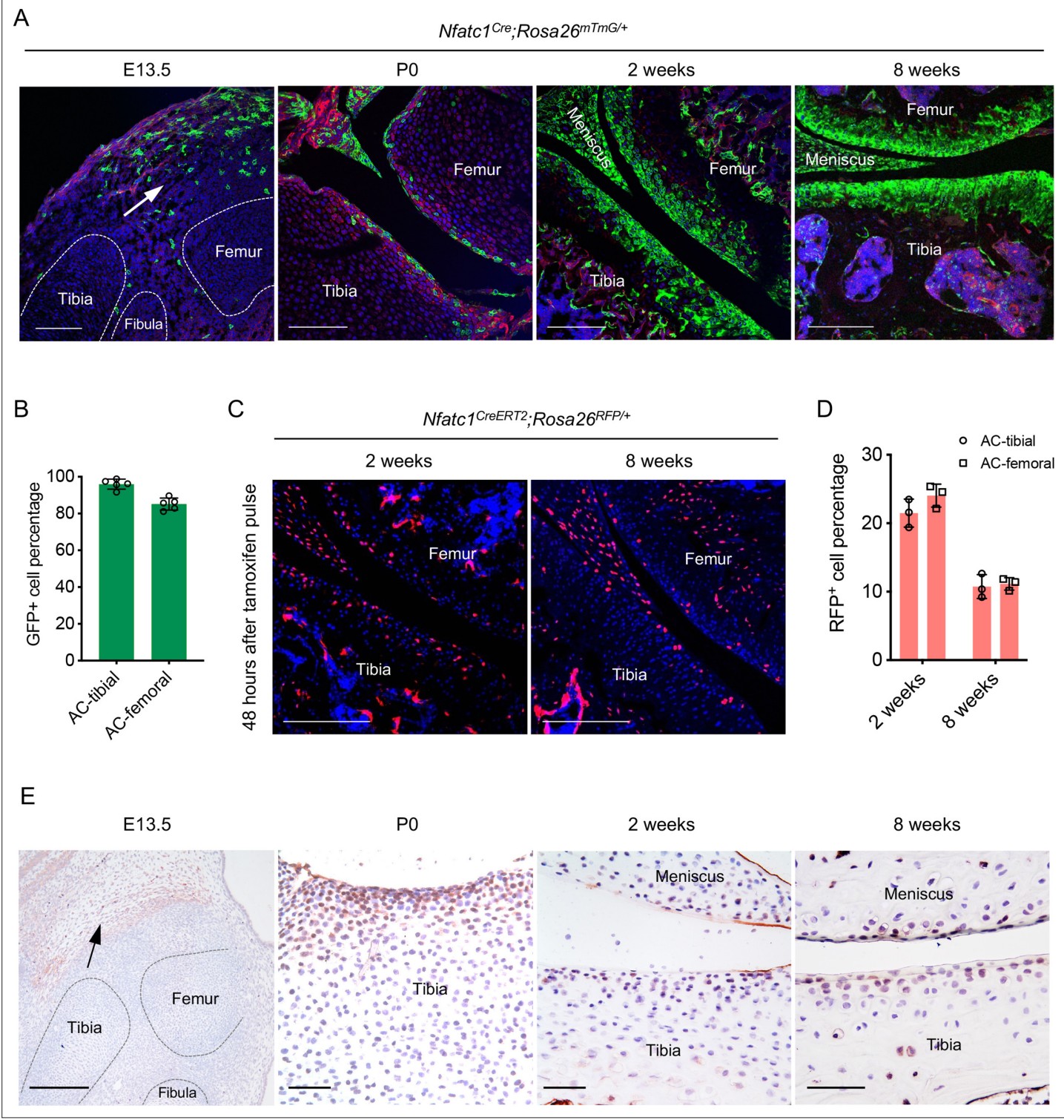

**Figure 1.** Articular cartilage is derived from NFATc1-expressing progenitors. (**A**) Confocal microscopy images showing the distribution of GFP$^+$ cells during articular cartilage development at the knee of *Nfatc1$^{Cre}$;Rosa26$^{mTmG/+}$* mice (n=5 animals for each age, two knee joints per animal). Arrow indicating the main location of GFP$^+$ cells at the knee at embryonic day 13.5 (E13.5). P0, postnatal day 0. (**B**) Quantification of GFP$^+$ cells in the articular cartilage of *Nfatc1$^{Cre}$;Rosa26$^{mTmG/+}$* mouse knee at 8 weeks of age (n=5 animals, one knee joint per animal). AC, articular cartilage. (**C**) Representative confocal images demonstrating the distribution of RFP$^+$ cells in the articular cartilage at 2 weeks and 8 weeks of age in *Nfatc1$^{CreERT2}$;Rosa26$^{RFP/+}$* mice 48 hrs after tamoxifen pulse for 5 consecutive days (n=3 mice for each age, two knee joints per animal). (**D**) Quantification of RFP$^+$ cells in the articular cartilage of *Nfatc1$^{CreERT2}$;Rosa26$^{RFP/+}$* mouse knee at 2 weeks and 8 weeks of age (n=3 mice for each age, one knee joint per animal).

*Figure 1 continued on next page*

*Figure 1 continued*

(**E**) Immunohistochemistry detecting the expression of NFATc1 during mouse articular cartilage development (n=3 mice for each age, two knee joints per animal). Data are mean ± SD of results from five or three animals; scale bars, 200 μm except for the right three images in (**E**), 50 μm.

The online version of this article includes the following source data and figure supplement(s) for figure 1:

**Source data 1.** Quantification data for GFP⁺ or RFP⁺ cells in articular cartilage.

**Figure supplement 1.** Track NFATc1-expressing progenitors during articular cartilage development.

scattered in the articular cartilage at 2 weeks of age accounting for about 22.75 ± 2.18% (n=3 mice) of all cells of articular cartilage, while most RFP⁺ cells were confined to the superficial layers of articular cartilage at 8 weeks of age (10.94 ± 1.26%, n=3) [*Figure 1(C and D)*]. Furthermore, the expression pattern of NFATc1 in mouse articular cartilage was verified by immunohistochemistry at E13.5, P0, 2 weeks, and 8 weeks of age [*Figure 1(E)* and *Figure 1—figure supplement 1B*]. Therefore, by mapping GFP and RFP expression in articular cartilage of *Nfatc1^Cre^;Rosa26^mTmG/+^* and *Nfatc1^CreERT2^;Rosa26^RFP/+^* mice respectively [*Figure 1—figure supplement 1C*], many GFP⁺ articular chondrocytes in *Nfatc1^Cre^;Rosa26^mTmG/+^* mice at 2 weeks and 8 weeks of age should be derived from *Nfatc1*-expressing progenitors and had lost the expression of NFATc1 with development.

Notably, there were no GFP⁺ chondrocytes in the primordium of growth-plate cartilage at E13.5 in both *Nfatc1^Cre^;Rosa26^mTmG/+^* and tamoxifen-induced *Nfatc1^CreERT2^;Rosa26^mTmG/+^* mice [*Figure 1(A)* and *Figure 1—figure supplement 1A*], suggesting that NFATc1-expressing cells do not generate the cartilaginous primordium of growth-plate. We did not detect GFP⁺ cells in articular cartilage in *Rosa26^mTmG/+^* control mice and *Nfatc1^CreERT2^;Rosa26^mTmG/+^* mice without tamoxifen induction [*Figure 1—figure supplement 1D*], suggesting that there was no Cre leakage in the articular cartilage in these two reporter mouse lines. Together, these results reveal that articular chondrocytes are derived from NFATc1-expressing progenitors and NFATc1 expression is diminished with articular cartilage development.

## Colony formation and multipotent differentiation of NFATc1-expressing progenitors

The lineage tracing data in *Nfatc1^Cre^* and *Nfatc1^CreERT2^* reporter mice suggest that NFATc1 characterizes articular cartilage progenitors. In this context, the fluorescence-labeled cells after tamoxifen-induced recombination in *Nfatc1^CreERT2^* reporter mice should be able to form in vivo cell clones in or next to the articular cartilage with development. To verify this assumption, we exploited the *Nfatc1^CreERT2^;Rosa26^mTmG/+^* double-fluorescence reporter mouse line and administered two dosages of tamoxifen to dams at P0 and P1, respectively. One week following the tamoxifen pulse, GFP⁺ cells were detected at the presumptive articular cartilage site at the mouse knee [*Figure 2(A)*]. Local GFP⁺ cell clusters with 3–6 cells each could be observed in articular cartilage by 2 weeks and 8 weeks of age [*Figure 2(A and B)*]. Notably, GFP⁺ cell clusters were also found in the meniscus, synovial lining, and ligament [*Figure 2(A and B)*], suggesting that NFATc1 also marks progenitor cells for joint tissues other than articular cartilage. Indeed, in *Nfatc1^Cre^;Rosa26^mTmG/+^* mice, GFP⁺ cells also formed the meniscus, synovial lining, ligament, and primordium of the patella at the knee [*Figure 1(A)* and *Figure 2(C)*].

To further characterize the colony formation capacity of NFATc1-expressing articular cartilage progenitors, we cultured and sorted GFP⁺ cells and their counterparts (GFP⁻ cells) from the knee of neonatal *Nfatc1^Cre^;Rosa26^mTmG/+^* mice [*Figure 3(A)*]. The ex vivo colony formation assay showed that GFP⁺ cells formed remarkably more numerous and larger cell clones in comparison with GFP⁻ cells when plated at the same cell densities and cultured for the same time period [*Figure 3(B)*]. A similar outcome could be observed even after five consecutive cell passages [*Figure 3—figure supplement 1A*]. Thus, NFATc1-expressing articular cartilage progenitors display a rigorous capacity for colony formation both in vivo and ex vivo.

To study the differentiation potentials of NFATc1-expressing articular cartilage progenitors, we put GFP⁺ and GFP⁻ cells under chondrogenic, osteogenic, and adipogenic differentiation conditions, respectively. Notably, GFP⁺ cells displayed a much higher potential to differentiate toward chondrocytes, osteoblasts, and adipocytes compared to GFP⁻ cells [*Figure 3(C–E)*]. A more striking difference was noticed under the context of chondrocyte differentiation: in the 3D cell-pellet culture model, GFP⁺ cells always grew into larger pellets as shown by the diameter of cell pellets and displayed a

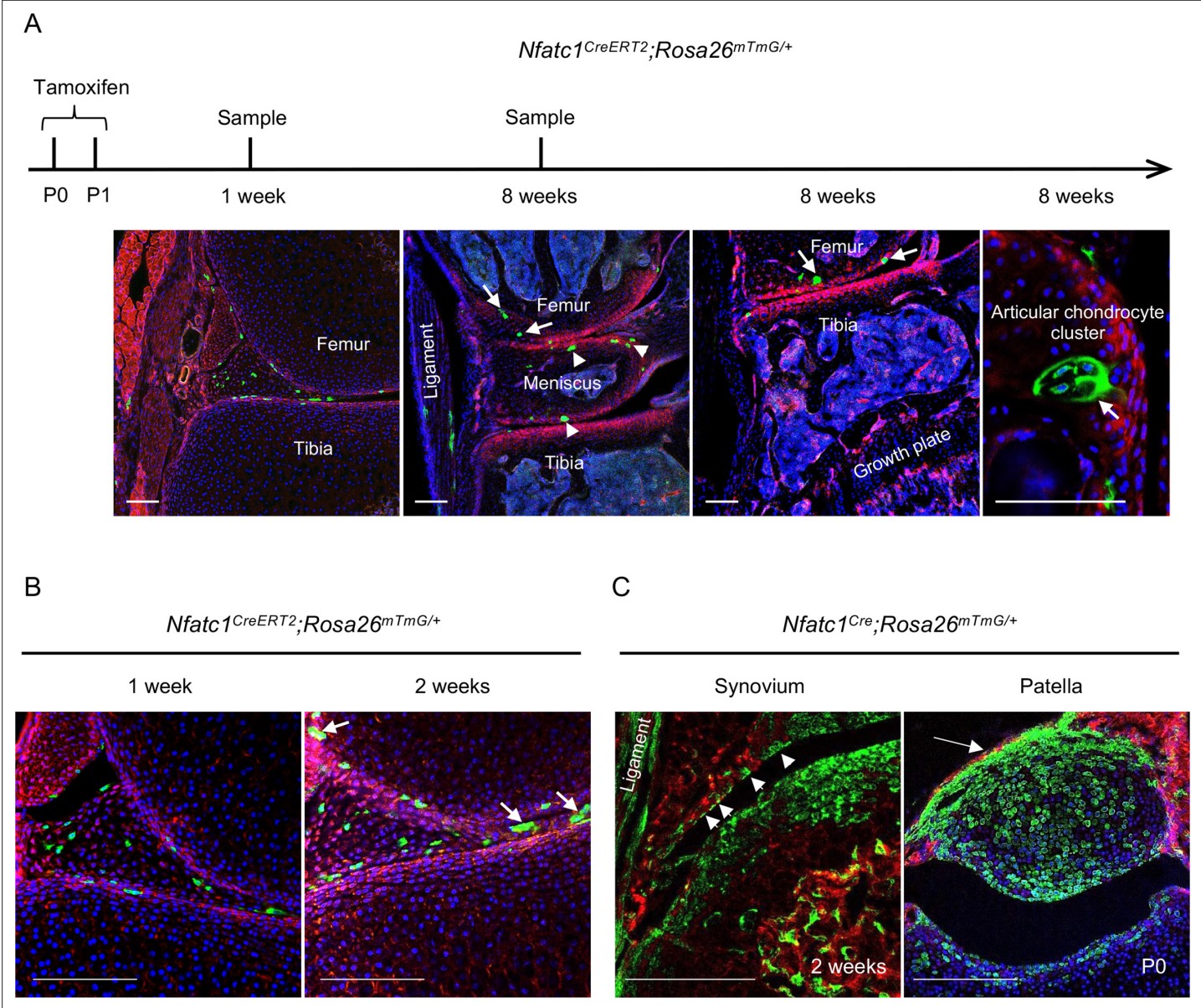

**Figure 2.** NFATc1-expressing progenitors form cell clusters with articular cartilage development and contribute to the meniscus and articular synovium formation. (**A**) Confocal microscopy images showing the distribution of GFP+ cells in the articular cartilage of *Nfatc1^CreERT2^;Rosa26^mTmG/+^* mouse knee at 1 week and 8 weeks after administering tamoxifen to dams at P0 and P1. The most left image showing GFP+ cells in articular tissues after 1 week of tamoxifen administration. The right three images demonstrating GFP+ cell clusters in the articular cartilage (arrows), meniscus (arrowheads), and ligament. (**B**) Representative confocal images displaying GFP+ cells or cell clusters (arrows) in the meniscus and articular synovium at 1 week or 2 weeks after tamoxifen administration to dams at P0 and P1. (**C**) Confocal microscopy images demonstrating that GFP+ cells contribute to the formation of the ligament, synovial lining (left image, arrowheads, 2 weeks of age), and the patella of the knee (right image, arrow, **P0**) in *Nfatc1^Cre^;Rosa26^mTmG/+^* mice. All images are representative of five mice at each time point or age, two knee joints per animal. Scale bars, 200 μm.

robust capacity of chondrocyte differentiation as shown by alcian blue staining and COL2A1 protein expression, while GFP- cells formed relatively small pellets and only displayed faint cartilage formation at the margin of cell pellets [***Figure 3(C)***], indicating that GFP+ cells have a more prominent capacity for proliferation and chondrogenesis compared to GFP- cells. Furthermore, when transplanted alongside with Matrigel matrix underneath the dorsal skin of SCID mice, these GFP+ cells differentiated and formed typical chondrocytes as well as chondrocyte clusters within 4 weeks, while the formation of chondrocytes was rarely observed when transplanting GFP- cells [***Figure 3(F)*** and ***Figure 3— figure supplement 1B***]. Notably, many chondrocyte clusters from GFP+ cells had formed hypertrophic

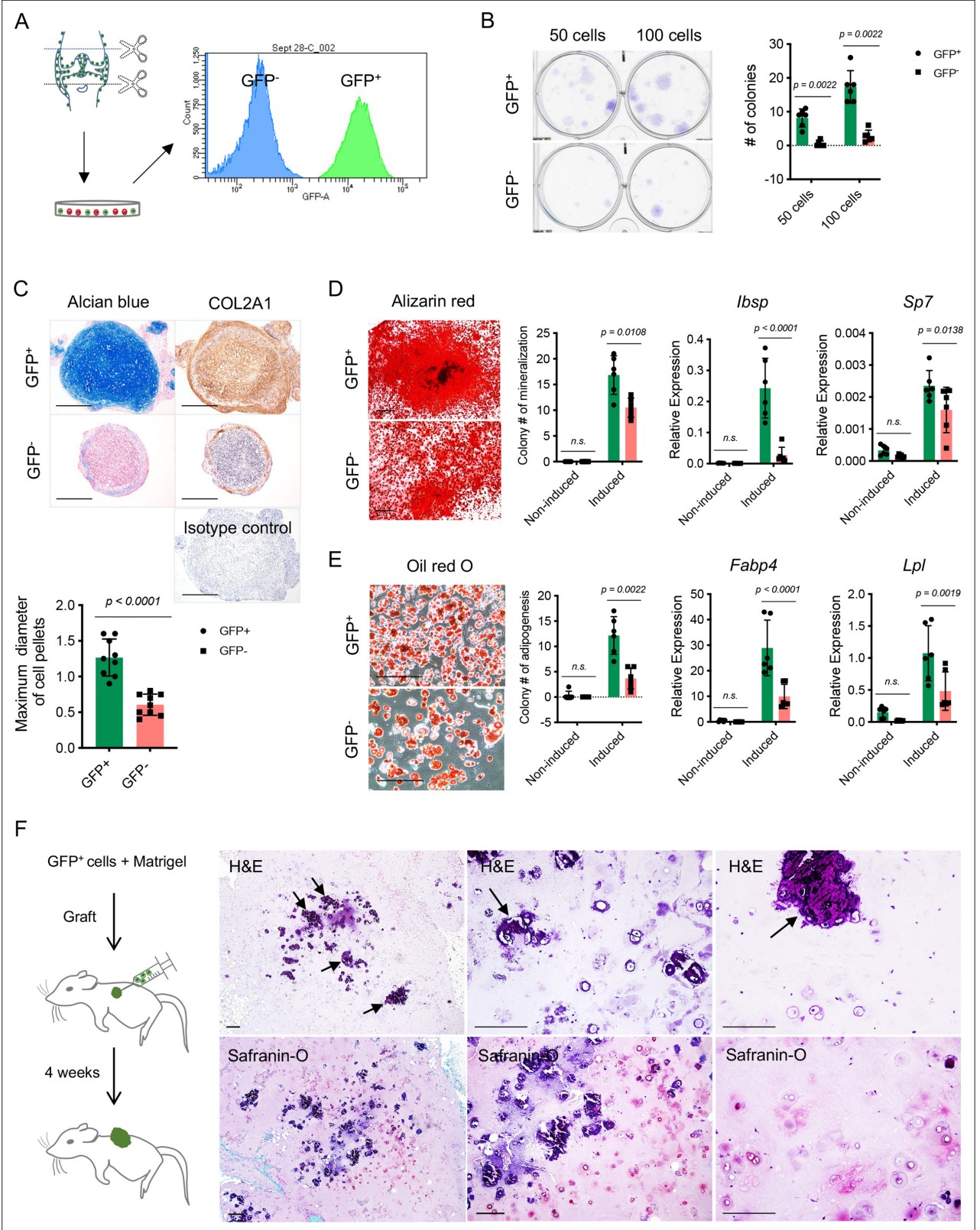

**Figure 3.** Colony formation and multipotent differentiation of *Nfatc1*-expressing progenitors. (**A**) Schematic diagram showing culturing and sorting GFP⁺ and GFP⁻ cells from the knee of neonatal *Nfatc1^Cre^;Rosa26^mTmG/+^* mice. (**B**) Colony formation assay of GFP⁺ and GFP⁻ cells with 50 or 100 cells plated in 6-well-plates and cultured for 2 weeks. n=6 with cells from three animals, two replicates for each, nonparametric Mann-Whitney test, experiment repeated twice. (**C**) Alcian blue staining and immunohistochemistry of COL2A1 showing the chondrogenic potential of GFP⁺ and GFP⁻ cell pellets after

*Figure 3 continued on next page*

*Figure 3 continued*

being cultured in the chondrogenic differentiation medium for 3 weeks. Isotype as a negative control for COL2A1 antibody. The maximum diameter of cell pellets reflecting the proliferative capacity of GFP⁺ and GFP⁻ cells. n=9 with cells from three animals, three replicates for each. (**D**) Alizarin red staining and gene expression analysis of *Ibsp* and *Sp7* demonstrating the osteogenic potential of GFP⁺ and GFP⁻ cells after being cultured in the osteogenic differentiation medium for 4 weeks. n=6 with cells from three animals, two replicates for each, nonparametric Mann-Whitney test for colony counting data, two-way ANOVA followed by Sidak's tests for gene expression data, experiments repeated twice. (**E**) Oil red O staining and gene expression analysis of *Fabp4* and *Lpl* displaying adipogenesis in GFP⁺ and GFP⁻ cells after being cultured in the adipogenic differentiation medium for 10 days. n=6 with cells from three animals, two replicates for each, nonparametric Mann-Whitney test for colony counting data, two-way ANOVA followed by Sidak's tests for gene expression data, experiments repeated twice. (**F**) Schematic illustration and histology respectively showing transplantation of GFP⁺ cells along with Matrigel matrix underneath the dorsal skin of severe combined immune-deficient mice and the formation of chondrocytes, chondrocyte clusters, and hypertrophic cartilage-like structure (arrows) 4 weeks later. Images are representative of six animals, with GFP⁻ cells as the control (results shown in ***Figure 3—figure supplement 1B***). All data are mean ± SD. Scale bars, 400 μm (**C**), 500 μm (**D**), 200 μm (**E, F**).

The online version of this article includes the following source data and figure supplement(s) for figure 3:

**Source data 1.** Data of colony numbers, cell pellet diameters, and qPCR.

**Figure supplement 1.** Colony formation assay and in vivo transplantation of GFP⁻ cells.

**Figure supplement 1—source data 1.** For ***Figure 3—figure supplement 1A*** Colony formation assay of GFP⁺ and GFP⁻ cells at P5 with 100 cells plated in 6-well-plates and cultured for 2 weeks (colony numbers).

cartilage-like tissue with a certain hardness, similar to the physiological process of articular cartilage development [***Figure 3(F)***].

Taken together, these results demonstrate the intrinsic capacities of colony formation and multipotent differentiation of NFATc1-expressing articular cartilage progenitors.

## Transcriptional profile of NFATc1-enriched articular cartilage progenitors

Next, we sought to dissect the molecular signature of NFATc1-enriched articular cartilage progenitors. In order to minimize the influence of differentiated cells in GFP⁺ and GFP⁻ cell populations, single-clone cells were sorted at the first passage (P1), amplified for one more passage, and subjected to transcriptome analysis at P2 [***Figure 4(A)***]. Bioinformatics analysis identified 117 high- and 168 low-expressing genes in GFP⁺ *vs.* GFP⁻ cells [***Figure 4—figure supplement 1A***]. High-expressing genes in GFP⁺ cells were mainly related to skeletal system development, cartilage development, or extracellular matrix component and organization [***Figure 4(B)*** and ***Figure 4—figure supplement 1B***]. Of note, these high-expressing genes in GFP⁺ cells included several previously reported articular cartilage progenitor cell markers *Decker, 2017*; *Bian et al., 2020*, such as *Osr2*, *Prg4*, *Postn*, *Col3a1*, *Gdf6*, and *Tgfbr2* [***Figure 4(C)***]. In contrast, enriched genes in GFP⁻ cells were mainly relevant to muscle cell development and differentiation [***Figure 4—figure supplement 1C***], suggesting that GFP⁻ single-clone cells could be skeletal muscle progenitors. Importantly, the characteristic molecular signature and enriched biological pathways in GFP⁺ cells were verified by the second transcriptome analysis using bulk primary GFP⁺ cells [***Figure 4—figure supplement 1D-F***].

Cell surface markers are important in identifying and sorting progenitor or stem cells. Transcriptome analyses showed that both GFP⁺ and GFP⁻ progenitors expressed several surface markers of cells of mesenchymal origin, including *Cd9*, *Sca1*, *Thy1*, *Cd73*, *Cd166*, *Cd200*, and *Cd51*, but not hematopoietic or endothelial markers *Cd11b*, *Cd45*, or *Cd31* (***Supplementary file 1***). When compared with GFP⁻ cells, GFP⁺ cells displayed higher expression of *Cd105*, *Cd10*, and *Cd13* and lower expression of *Cd146*, *Cd29*, and *Cd151* [***Figure 4(D)***]. The expression of CD105, CD9, SCA1, CD166, CD200, CD11B, CD45, and CD31 were further verified by flow cytometry in GFP⁺ and GFP⁻ cells [***Figure 4(E and F)*** and ***Figure 4—figure supplement 2***]. Combined, these data identify a set of genes preferentially expressed in NFATc1-expressing articular cartilage progenitors and provide a perspective to understand the transcriptional signature of these progenitors.

## NFATc1 negatively regulates articular chondrocyte differentiation

The function of NFATc1 in articular cartilage progenitors remains unclear. As aforementioned, the lineage tracing of *Nfatc1*-expressing cells showed that most articular chondrocytes were GFP⁺ in *Nfatc1^Cre^;Rosa26^mTmG/+^* mice at 8 weeks of age, but the real-time expression of NFATc1 was confined to the superficial layers of articular cartilage as shown by RFP expression after tamoxifen pulse in

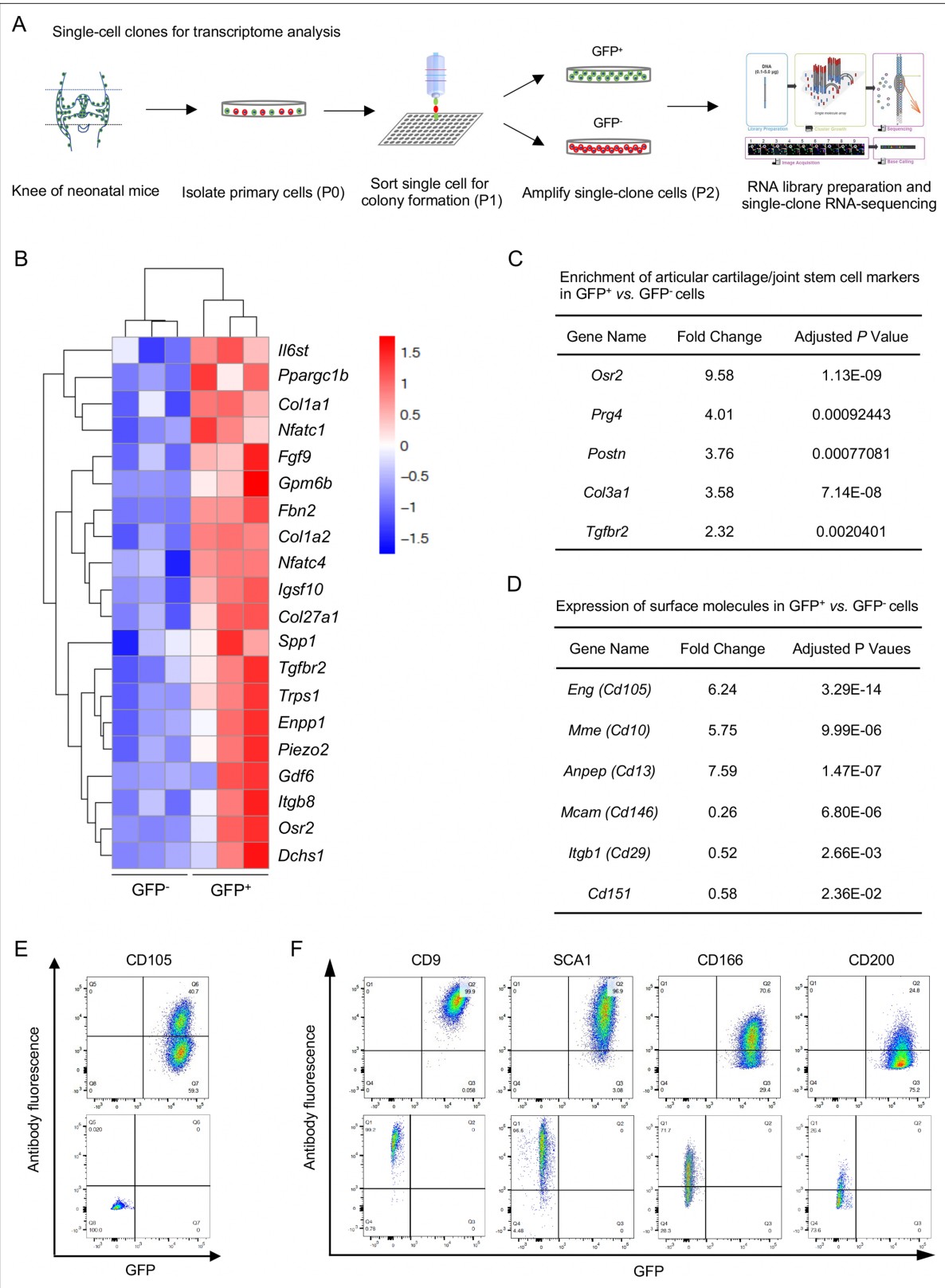

**Figure 4.** Transcriptional profile of *Nfatc1*-expressing articular cartilage progenitors. (**A**) Schematic diagram showing the process of sorting single-clone cells for RNA-sequencing. (**B**) Cluster heatmap displaying 20 high-expressing genes associated with articular cartilage development in GFP⁺ *vs.* GFP⁻ cells. Color descending from red to blue indicates log10(FPKM +1) from large to small. n=3 with cells from three animals in each group. (**C**) Transcriptome analysis revealing the enrichment of previously reported articular cartilage progenitor marker genes *Osr2*, *Prg4*, *Postn*, *Col3a1*, and

*Figure 4 continued on next page*

*Figure 4 continued*

*Tgfbr2* in GFP⁺ relative to GFP⁻ cells. (**D**) Transcriptome analysis identifying high expression of *Cd105*, *Cd10*, and *Cd13* and low expression of *Cd146*, *Cd29*, and *Cd151* in GFP⁺ *vs*. GFP⁻ cells. (**E**) Flow cytometry verifying the expression of CD105 in GFP⁺ relative to GFP⁻ cells. (**F**) Flow cytometry showing the expression of cell surface molecules CD9, SCA1, CD166, and CD200 in GFP⁺ and GFP⁻ cells. Representative results of cells from three mice, experiment repeated twice.

The online version of this article includes the following figure supplement(s) for figure 4:

**Figure supplement 1.** Transcriptional profile of *Nfatc1*-expressing articular cartilage progenitors.

**Figure supplement 2.** Flow cytometry determining the expression of cell surface markers.

*Nfatc1*$^{CreERT2}$;*Rosa26*$^{RFP/+}$ mice [**Figure 1—figure supplement 1C**]. These results indicate that NFATc1 expression was diminished with articular chondrocyte differentiation. Consistently, the diminishment of *Nfatc1* expression was also detected in ex vivo chondrogenesis in *Nfatc1*-expressing articular cartilage progenitors [**Figure 5(A)**].

Next, we wondered whether suppressing NFATc1 expression in articular cartilage progenitors is sufficient to induce chondrocyte differentiation. To address this, we deleted NFATc1 expression in GFP⁺ articular cartilage progenitors by CRISPR/CAS9 technique [**Figure 5—figure supplement 1A**]. Strikingly, the deletion of NFATc1 expression was enough to induce chondrocyte differentiation without adding the chondrogenic medium as shown by alcian blue staining and induced expression of *Acan*, *Col2a1*, and *Col10a1* [**Figure 5(B)**]. In the meantime, overexpression of NFATc1 in GFP⁺ progenitor cells inhibited chondrocyte differentiation [**Figure 5(C)** and **Figure 5—figure supplement 1A**].

To verify the function of NFATc1 in the regulation of articular chondrogenesis in vivo, we conditionally deleted *Nfatc1* in *Prrx1*-expressing limb mesenchymal progenitors by crossing *Prrx1*$^{Cre}$ mice with *Nfatc1*$^{fl/fl}$ mice. The staining for articular cartilage was markedly strengthened in *Prrx1*$^{Cre}$;*Nfatc1*$^{fl/fl}$ (homozygous) *vs*. *Prrx1*$^{Cre}$;*Nfatc1*$^{fl/+}$ (heterozygous) mice [**Figure 5(D and E)**]. Consistently, the expression of *Acan*, *Col2a1*, and *Col10a1* genes was significantly upregulated in the articular cartilage of *Prrx1*$^{Cre}$;*Nfatc1*$^{fl/fl}$ mice [**Figure 5(F)**]. Notably, the articular cartilage consisted of mostly round, large chondrocytes in *Prrx1*$^{Cre}$;*Nfatc1*$^{fl/fl}$ mice, instead of the normal transitional alignment from the superficial layer to the deep layers [**Figure 5(E)**]. The volume changes of articular chondrocytes resulted in an increased thickness of articular cartilage in *Prrx1*$^{Cre}$;*Nfatc1*$^{fl/fl}$ *vs*. *Prrx1*$^{Cre}$;*Nfatc1*$^{fl/+}$ mice [**Figure 5(E)**]. These results indicate that the in vivo deletion of *Nfatc1* in limb mesenchymal progenitors promotes articular chondrocyte differentiation as well. Note that a previous study showed that *Nfatc1* deletion in *Col2a1*-expressing cells (*Nfatc1*$^{Col2a1}$ mice) does not affect articular cartilage integration or osteoarthritis progression induced by destabilization of the medial meniscus **Greenblatt et al., 2013**, suggesting that NFATc1 is dispensable in *Col2a1*-expressing differentiated articular chondrocytes. Therefore, NFATc1 may primarily function in articular cartilage progenitors to regulate articular chondrocyte differentiation.

Based on gene expression changes after deleting or overexpressing NFATc1, *Col2a1* turned out to be one of the most significantly changed chondrocyte-related genes we examined [**Figure 5(B, C and F)** and reference **Ge et al., 2016**]. To understand the mechanism of NFATc1 regulating articular chondrocyte differentiation, we performed computational screening and identified a total of 38 potential NFAT binding sites across the upstream 6 k base pairs of exon 1 and the intron 1 of mouse *Col2a1* gene [**Figure 5(G)**]. To narrow down the NFATc1 binding sites in the regulatory region of the *Col2a1* gene, we performed a FIMO analysis **Grant et al., 2011** and identified a NFATc1 motif that locates in mouse chr15: 98004609–98004620 and overlaps with the *Col2a1* promoter region [**Figure 5—figure supplement 1B**]. Furthermore, we verified the binding of NFATc1 to this site using the technique of cleavage under targets and release using nuclease (CUT&RUN) [**Figure 5(H)**]. These results show that NFATc1 might play a role in regulating *Col2a1* gene transcription. Indeed, the luciferase analysis showed that silence or overexpression of NFATc1 markedly upregulated or downregulated the transcriptional activity of *Col2a1*, respectively [**Figure 5(I)**]. Therefore, NFATc1 negatively determines articular chondrocyte differentiation at least partly through regulating the transcription of the *Col2a1* gene.

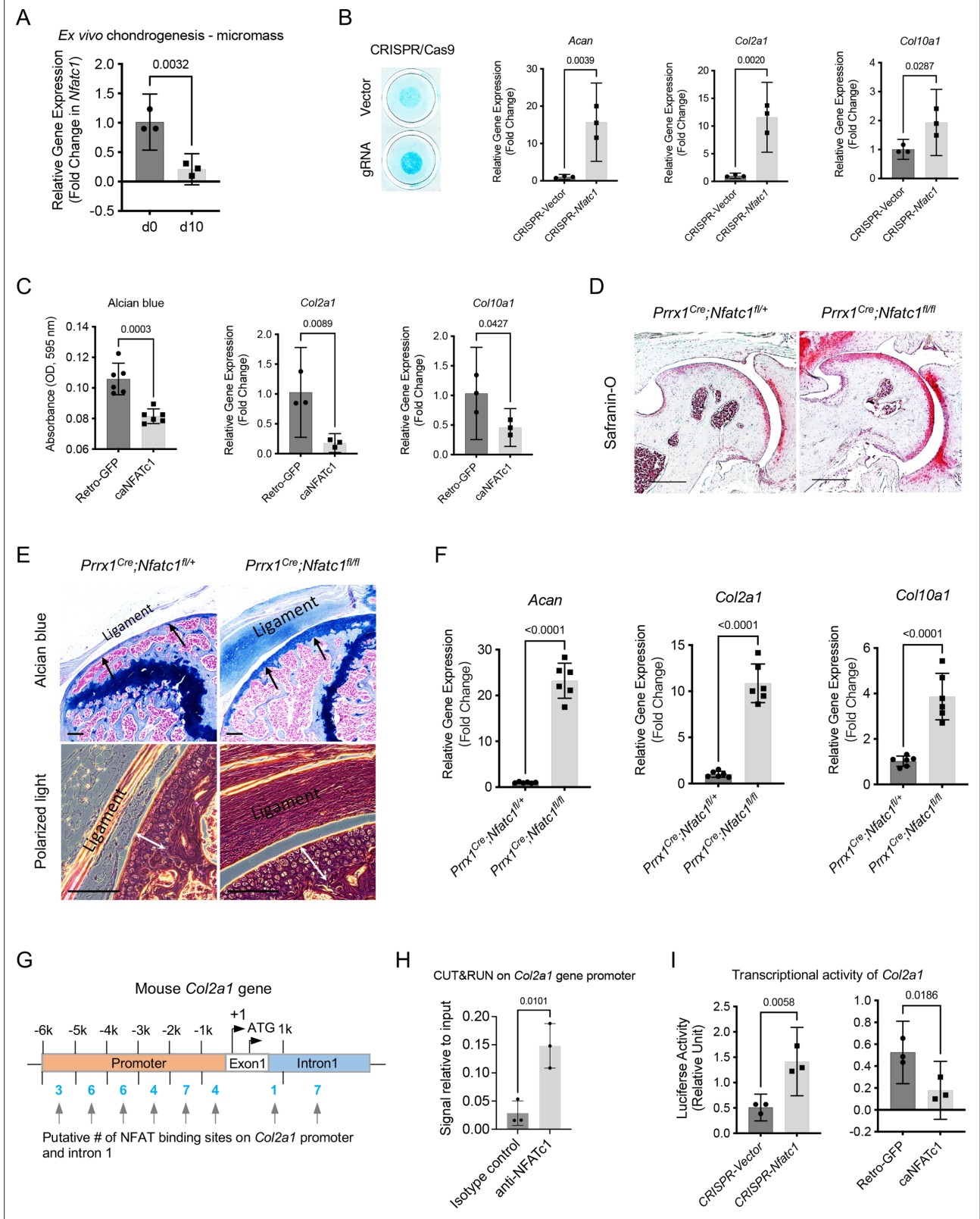

**Figure 5.** NFATc1 negatively determines articular chondrocyte differentiation through regulating *Col2a1* gene transcription. (**A**) Gene expression analysis displaying the change of *Nfatc1* expression in GFP⁺ cell-micromass cultured in the chondrogenic medium for 10 days. n=3, experiment repeated three times with cells from three mice. (**B**) Alcian blue staining and gene expression analysis of *Acan*, *Col2a1*, and *Col10a1* after deleting *Nfatc1* by CRISPR/CAS9 technique in ex vivo micromass culture of GFP⁺ cells from neonatal *Nfatc1^Cre^;Rosa26^mTmG/+^* mice (day 7, without chondrogenic

*Figure 5 continued on next page*

*Figure 5 continued*

induction, n=3). Experiment repeated three times with cells from three animals. (**C**) Quantification of alcian blue staining and gene expression analysis of *Col2a1* and *Col10a1* showing decreased chondrogenesis after overexpressing NFATc1 in GFP⁺ cells by infecting a caNFATc1 retrovirus structure. For alcian blue staining, n=6 with cells from three animals, two replicates for each; for gene expression analysis, n=3, experiment repeated twice with cells from two animals. (**D**) Safranin O staining demonstrating enhanced articular cartilage staining in the hip of *Prrx1^Cre^;Nfatc1^fl/fl^ vs. Prrx1^Cre^;Nfatc1^fl/+^* mice at 12 weeks of age. Representative images from five animals in each group were displayed. (**E**) Representative images of alcian blue staining and polarized light on H&E staining manifesting increased staining (arrows) and thickness of articular cartilage (double arrows) in the knee of *Prrx1^Cre^;Nfatc1^fl/fl^* relative to *Prrx1^Cre^;Nfatc1^fl/+^* mice at 16 weeks of age. n=5 animals for each group. (**F**) Quantitative PCR determining the expression of *Acan*, *Col2a1*, and *Col10a1* genes in articular cartilage of *Prrx1^Cre^;Nfatc1^fl/fl^* relative to *Prrx1^Cre^;Nfatc1^fl/+^* mice at 8 weeks of age. n=6 animals for each group. (**G**) Computational screening of NFAT binding sites on mouse *Col2a1* promoter and intron 1 sequences by PROMO software recognizing 38 putative NFAT binding sites across –6 k bp of the promoter and intron 1. The transcriptional starting site is counted as +1. Location is given in bp relative to the transcriptional starting site. (**H**) Cleavage under targets and release using nuclease (CUT&RUN)-qPCR showing the binding of NFATc1 to the promoter region of mouse *Col2a1* (chr15: 98004609–98004620, mm10). (**I**) Luciferase assay of transcriptional activity of *Col2a1* after deleting or overexpressing NFATc1 in ATDC5 cells. n=3, experiment repeated three times. All data shown as mean ± SD, two-tailed Student's *t*-test performed. Scale bar, 200 μm (**D**), 100 μm (**E**).

The online version of this article includes the following source data and figure supplement(s) for figure 5:

**Source data 1.** Data of qPCR and luciferase assay.

**Figure supplement 1.** Western blot determining NFATc1 expression and NFATc1 binding site at the *Col2a1* gene promoter by FIMO analysis.

**Figure supplement 1—source data 1.** Raw uncropped blots.

## Discussion

For quite a long period, the interzone cells have been considered as the origin of articular chondrocytes *Chijimatsu and Saito, 2019*; *Rux et al., 2019*. Also, progenitor cells in the flanking mesenchyme surrounding the joint interzone were found to migrate into the interzone region and form articular cartilage *Shwartz et al., 2016*; *Decker et al., 2017*; *Niedermaier et al., 2005*; *Koyama et al., 2007*. The current obscurity in identifying the precise origin of articular chondrocytes could be attributed to the lack of a specific molecular marker to distinguish cells in the flanking region from interzone cells. From this perspective, the limited expression of NFATc1 in the flanking region at the primary stage of mouse knee development [E13.5, *Figure 1(A and E)* and *Figure 1—figure supplement 1A*] provides a unique opportunity to track progenitors in the flanking region during articular cartilage development. The progressive pattern of NFATc1-expressing cells contributing to articular cartilage formation (*Figures 1–2*) suggests that NFATc1-expressing progenitors in the flanking region may represent the origin of articular chondrocytes. Further studies into the spatiotemporal roadmap of NFATc1-expressing progenitor cell development are essential to elucidate the landscape of articular cartilage formation. Notably, these results also demonstrate that NFATc1 can constantly track articular cartilage progenitors throughout embryonic development and postnatal growth.

Transcriptome analyses in both single-clone and bulk primary cells reveal that NFATc1-expressing articular cartilage cells enrich several previously identified articular cartilage progenitor cell markers including *Osr2*, *Prg4*, *Postn*, *Col3a1*, and *Gdf6* [*Figure 4(C)*; *Decker, 2017*; *Bian et al., 2020*], therefore advocating their identity as articular cartilage progenitors. In addition, cell surface molecules including CD105, CD13, CD10, CD9, SCA1, and CD166 can be considered as complementary markers for identifying and screening articular cartilage progenitors. Most impressively, NFATc1-expressing articular cartilage progenitors enrich a set of genes like *Fbn2*, *Piezo2*, *Dchs1*, *Enpp1*, *Gdf6*, *Fgf9*, *Trps1*, *Col27a1*, *Tgfbr2*, *Col1a1*, and *Col1A2* [*Figure 4(B)*], whose mutations have been linked to a diverse range of human musculoskeletal disorders (https://www.omim.org). These findings suggest that the dysfunction of articular cartilage progenitors may underlie these human musculoskeletal disorders. Of note, we did not detect the expression of *Gdf5* in articular cartilage progenitors from neonatal mice, which coincides with previous reports that *Gdf5* is greatly diminished with synovial joint development and undetectable in the articular cartilage of neonatal mice *Decker, 2017*; *Francis-West et al., 1999*; *Merino et al., 1999*. However, the enriched expression of *Gdf6* may provide a complementary role to *Gdf5* [*Figure 4(B)*; *Settle et al., 2003*]. Furthermore, based on the distribution of GDF5⁺ progenitors during synovial joint development and their contribution to the formation of most articular chondrocytes in previous reports *Decker, 2017*; *Decker et al., 2017*, NFATc1-expressing articular cartilage progenitors are probably a subset of GDF5⁺ progenitors. Together, these results provide important insights into the molecular signature of articular cartilage progenitors.

Our results show that suppression of NFATc1 is sufficient to induce articular chondrogenesis both ex vivo and in vivo [*Figure 5(B and D–F)*] and thus NFATc1 could represent a core negative transcriptional regulator of articular chondrocyte differentiation. Notably, this role of NFATc1 in regulating chondrocyte differentiation is confined to the articular cartilage because NFATc1 does not express in growth-plate progenitors or chondrocytes [*Figure 1(A and E)* and *Figure 1—figure supplement 1A*]. Consistent with the inhibitory role of NFATc1 in articular chondrocyte differentiation, previous in vitro studies in murine chondrogenic cell line ATDC5 cells and human epiphyseal chondrocytes also showed that NFATc1 suppresses chondrocyte differentiation *Zanotti and Canalis, 2013*.

Notably, a protective role of NFATc1 in osteoarthritic cartilage has been suggested previously, which is based on the decrease of NFATc1 expression in lesional osteoarthritic cartilage in human patients, as well as the severe articular cartilage deterioration in mice with conditional deletion of *Nfatc1* driven by the Collagen 2 promoter Cre with a background of *Nfatc2* deficiency (*Nfatc1$^{Col2a1}$;Nfatc2$^{-/-}$* mice) *Greenblatt et al., 2013*; *Beier, 2014*. While the *Nfatc1$^{Col2a1}$;Nfatc2$^{-/-}$* mice indeed develop osteoarthritic phenotypes, we did not find the typic osteoarthritic phenotype in the follow-up study in mice with deletion of *Nfatc1* driven by tamoxifen-induced *Aggrecan$^{CreERT2}$* with the background of *Nfatc2* deficiency (*Nfatc1$^{AggrecanCreERT2}$;Nfatc2$^{-/-}$* mice) after administering tamoxifen at different ages. Instead, these animals develop obvious osteochondroma-like lesions at the entheseal site and within ligaments around the joint *Ge et al., 2016*. Since *Col2a1* is also expressed in perichondrial precursors *Akiyama et al., 2005*; *Ono et al., 2014*, the accelerated osteoarthritis phenotype in *Nfatc1$^{Col2a1}$;Nfatc2$^{-/-}$* mice could be secondary to the osteochondroma phenotype in these animals, instead of a direct beneficial role of NFATc1 in articular cartilage.

A recent study by Atsuta et al. using limb bud mesenchymal cells showed that the ectopic expression of NFATc1 seems to promote chondrocyte differentiation as shown by the alcian blue staining in micromass-cultured cells *Atsuta et al., 2019*). While additional experiments are necessary to confirm chondrocyte differentiation at the molecular level, overexpression of NFATc1 in cells from the whole limb bud might not reflect the physiological function of NFATc1 in chondrogenesis because NFATc1 expression is highly confined to articular and perichondrial progenitors during skeletal development according to the lineage tracing data in *Nfatc1$^{Cre}$* and *Nfatc1$^{CreERT2}$* reporter mice (*Figures 1 and 2* and reference *Ge et al., 2016*). Therefore, this disparity could also be due to the different cell types utilized in these two studies.

The mechanism of regulating NFATc1 expression during articular cartilage development remains unclear. A previous study showed that Notch signaling suppresses NFATc1 expression in ATDC5 cells and primary chondrocytes *Zanotti and Canalis, 2013*. Furthermore, the Notch signaling can be activated by mechanical loading in mandibular condylar chondrocytes and bone marrow stromal cells *Ziouti et al., 2019*; *Yan et al., 2021*. Therefore, it is possible that mechanical loading and Notch signaling act upstream of NFATc1 expression during articular cartilage development. Further studies need to verify this speculation and explore other biochemical, biophysical, and epigenetic pathways regulating NFATc1 expression during articular chondrocyte differentiation. In addition, our transcriptome data show that *Nfatc4* is also enriched in articular cartilage progenitors. The function of NFATc4 and its relationship with NFATc1 in regulating articular cartilage formation needs further investigation.

In summary, we have unveiled that NFATc1-expressing progenitors generate articular but not growth-plate chondrocytes during development and identified NFATc1 as a critical negative transcriptional regulator of articular chondrocyte differentiation. Given the importance of NFATc1 in articular chondrogenesis, modulating NFAT signaling in skeletal progenitors may represent a novel, precise strategy for articular cartilage regeneration and treating cartilaginous diseases.

## Limitations of the study

There are some important limitations to our present study. Firstly, the GFP$^+$ cells we used in this study might contain some cells not expressing NFATc1 but derived from NFATc1-expressing precursors. Given the prominent progenitor cell properties of the GFP$^+$ cell population, if these derived cells also display characteristics of progenitor cells, their precursors (NFATc1-expressing cells) will represent a higher hierarchy of progenitors or stem cells, which yet support our conclusions. In fact, NFATc1 is positively expressed in the incipient articular cartilage of neonatal mice [Fig. 1(E)] and these NFATc1-expressing cells can form local cell clusters with articular cartilage development [Fig. 2(A)], indicating that NFATc1 also marks articular cartilage progenitors in neonatal mice. Secondly, these GFP$^+$ cells

might include cells from other articular tissues (e.g. meniscus, articular synovium, and ligament). Since all articular tissues are derived from progenitor cells sharing the same molecular markers *Koyama et al., 2008*; *Shwartz et al., 2016* and currently there are no specific markers to distinguish progenitors for these different articular tissues, it was challenging to explicitly isolate articular cartilage progenitor cells at the early stages of development. Future studies will be necessary to dissect the developmental hierarchy of NFATc1-expressing articular cartilage progenitor cells and elucidate the mechanisms determining their differentiation to meniscal, synovial, or ligament cells versus articular chondrocytes. Thirdly, the detailed molecular mechanism of NFATc1 regulating articular chondrocyte differentiation needs further exploration. Future studies combining techniques like ChIP, CUT&RUN, or CUT&Tag with high-throughput DNA sequencing will be able to unveil the transcriptional landscape of NFATc1-regulated articular chondrocyte differentiation. Lastly, the origin of NFATc1⁺ articular cartilage progenitors should be further explored, which will be critical to better understand the basic mechanism of articular cartilage development and leverage it for articular cartilage regeneration.

# Materials and methods

## Key resources table

| Reagent type (species) or resource | Designation | Source or reference | Identifiers | Additional information |
|---|---|---|---|---|
| Genetic reagent (*Mus. musculus*) | *Nfatc1$^{tm1.1(cre)Bz}$*; *Nfatc1$^{Cre}$* | PMID:23178125 | RRID:MGI:5471107 | Dr. Bin Zhou (Albert Einstein College of Medicine) |
| Genetic reagent (*M. musculus*) | *Nfatc1$^{tm1.1(cre/ERT2)Bzsh}$*; *Nfatc1$^{CreERT2}$* | PMID:24994653 | RRID:MGI:5637438 | Dr. Bin Zhou (Shanghai Institutes for Biological Sciences) |
| Genetic reagent (*M. musculus*) | B6(Cg)-*Nfatc1$^{tm3Glm}$*/AoaJ; *Nfatc1$^{fl}$* | The Jackson Laboratory | Strain #:022786 RRID:IMSR_JAX:022786 | |
| Genetic reagent (*M. musculus*) | B6.Cg-Tg(Prrx1-cre)1Cjt/J; *Prrx1$^{Cre}$* | The Jackson Laboratory | Strain #:005584 RRID:IMSR_JAX:005584 | |
| Genetic reagent (*M. musculus*) | B6.129(Cg)-Gt(ROSA)26 *Sor$^{tm4(ACTB-tdTomato,-EGFP)Luo}$*; *Rosa26$^{mTmG}$*; mTmG | The Jackson Laboratory | Strain #:007676 RRID:IMSR_JAX:007676 | |
| Genetic reagent (*M. musculus*) | B6.Cg-Gt(ROSA)26Sort$^{m9(CAG-tdTomato)Hze}$; *Rosa26$^{RFP}$*; Ai9 | The Jackson Laboratory | Strain #:007909 RRID:IMSR_JAX:007909 | |
| Genetic reagent (*M. musculus*) | CB17.Cg-*Prkdc$^{scid}$Lyst$^{bg-J}$*/Crl; SCID/ Beige mice | Beijing Vital River Laboratory Animal Technology Co., Ltd | Strain #:405 | |
| Antibody | anti-Collagen Type II (mouse monoclonal) | Merck (Millipore) | Cat#:MAB8887 RRID:AB_2260779 | (1:500) |
| Antibody | anti-NFATc1 antibody (mouse monoclonal) | Santa Cruz Biotechnology | Cat#:sc-7294 RRID:AB_2152503 | IHC (1:200); Western blotting (1:1000) |
| Antibody | Allophycocyanin (APC) anti-mouse CD45 (rat monoclonal) | Thermo Fisher Scientific | Cat#:MCD4505 RRID:AB_10376146 | (1:100) |
| Antibody | APC anti-mouse CD166 (rat monoclonal) | Thermo Fisher Scientific | Cat#:17-1661-82 RRID:AB_2573170 | (1:100) |
| Antibody | APC anti-mouse CD31 (rat monoclonal) | BioLegend | Cat#:102410 RRID:AB_312905 | (1:100) |
| Antibody | APC anti-mouse CD200 (rat monoclonal) | BioLegend | Cat#:123810 RRID:AB_10900447 | (1:100) |
| Antibody | APC anti-mouse SCA1 (rat monoclonal) | BioLegend | Cat#:122511 RRID:AB_756196 | (1:100) |
| Antibody | APC anti-mouse CD105 (rat monoclonal) | BioLegend | Cat#:120414 RRID:AB_2277914 | (1:100) |

*Continued on next page*

*Continued*

| Reagent type (species) or resource | Designation | Source or reference | Identifiers | Additional information |
|---|---|---|---|---|
| Antibody | APC Rat IgG2a, κ isotype control antibody (rat monoclonal) | BioLegend | Cat#:400512 RRID:AB_2814702 | (1:100) |
| Antibody | APC Rat IgG2b, κ isotype control antibody (rat monoclonal) | BioLegend | Cat#:400612 RRID:AB_326556 | (1:100) |
| Antibody | APC Rat IgG1, κ isotype control (rat monoclonal) | BioLegend | Cat#:400412 RRID:AB_326518 | (1:100) |
| Antibody | Alexa Fluor 647 anti-mouse CD9 (rat monoclonal) | BioLegend | Cat#:124810 RRID:AB_2076037 | (1:200) |
| Antibody | Alexa Fluor 647 Rat IgG2a, κ isotype control antibody (rat monoclonal) | BioLegend | Cat#:400526 RRID:AB_2864284 | (1:200) |
| Antibody | Go-ChIP-Grade purified anti-NFATc1 antibody (mouse monoclonal) | BioLegend | Cat#:649608 RRID:AB_2721596 | (10 µg/ml) |
| Antibody | Purified mouse IgG1, κ isotype ctrl antibody (mouse monoclonal) | BioLegend | Cat#:400102 RRID:AB_2891079 | CUT&RUN (10 µg/ml); IHC (1:500) |
| Antibody | anti-GAPDH (rabbit monoclonal) | Cell Signaling Technology | Cat#:2118 RRID:AB_561053 | (1:2000) |
| Chemical compound, drug | Tamoxifen | Sigma-Aldrich | Cat#:T5648 | Adult mice (1 mg/10 g); 2-week-old or dam mice (0.5 mg/10 g) |
| Cell line (*M. musculus*) | ATDC5 cells (mouse 129 teratocarcinoma-derived osteochondral progenitors) | Merck (Sigma-Aldrich) | Cat#:99072806 RRID:CVCL_3894 | Authenticated by chondrogenic differentiation; tested negative for mycoplasma |
| Sequence-based reagent | sgRNA targeting mouse *Nfatc1* exon 3 | This paper | | TACGAGCTTCGGATCGAGGT |
| Recombinant DNA reagent | pMSCV-caNFATc1; pMSCV-GFP | PMID:18243104 | | |
| Recombinant DNA reagent | pGL2B-COL2-6.5E309 | Other | | Dr. Mary Goldring |
| Transfected construct (mammalian) | gag/pol | Addgene | Cat#:14887 RRID:Addgene_14887 | |
| Transfected construct (mammalian) | VSV.G | Addgene | Cat#:14888 RRID:Addgene_14888 | |
| Transfected construct (mammalian) | pCMV-dR8.2 dvpr | Addgene | Cat#:8455 RRID:Addgene_8455 | |
| Transfected construct (mammalian) | pCMV-VSV-G | Addgene | Cat#:8454 RRID:Addgene_8454 | |
| Commercial assay or kit | M.O.M. Immunodetection Kit | Vector Laboratories | Cat#:BMK-2202 RRID:AB_2336833 | |
| Commercial assay or kit | CUT&RUN Assay Kit | Vazyme | Cat#:S702 | |
| Commercial assay or kit | Dual-Luciferase Reporter Assay System | Promega | Cat#:E1910 | |
| Software | FlowJo | BD Biosciences | RRID:SCR_008520 | Version 10.8.1 |
| Software | GraphPad Prism | GraphPad Prism (https://graphpad.com) | RRID:SCR_015807 | Version 9.2.1 |

## Mouse lines

The *Nfatc1*[tm1.1(cre)Bz] (*Nfatc1*[Cre]) **Wu et al., 2012** strain and *Nfatc1*[tm1.1(cre/ERT2)Bzsh] (*Nfatc1*[CreERT2]) **Tian et al., 2014** strain were generous gifts from Dr. Bin Zhou (Albert Einstein College of Medicine) and Dr. Bin Zhou (Shanghai Institutes for Biological Sciences, Chinese Academy of Sciences), respectively. The B6(Cg)-*Nfatc1*[tm3Glm]/AoaJ (*Nfatc1*[fl]) **Aliprantis et al., 2008**, B6.Cg-Tg(*Prrx1*[Cre])1Cjt/J (*Prrx1*[Cre]) **Logan et al., 2002**, Gt(ROSA)26 Sor[tm4(ACTB-tdTomato,-EGFP)Luo] (*Rosa26*[mTmG]) **Muzumdar et al., 2007**, and Gt(ROSA)26Sor[tm9(CAG-tdTomato)Hze] (*Rosa26*[RFP]) **Madisen et al., 2010** mice were obtained from the Jackson Laboratory. Severe combined immune deficient (SCID) beige mice were acquired from Beijing Vital River Laboratory Animal Technology Co., Ltd. All mice were housed under the standard barrier facility on a 12 hr light/dark cycle with ad libitum access to water and regular chow. All animal studies followed the recommendations in the Guide for the Care and Use of Laboratory Animals of the U.S. National Institutes of Health and were approved by Institutional Animal Care and Use Committee at Capital Medical University (protocol #: AEEI-2022–036). Animals were randomly assigned numbers and evaluated blindly to experimental conditions.

Tamoxifen (T5648, Sigma-Aldrich) was administered by intraperitoneal injection at 1 mg/10 g body weight for adult mice and 0.5 mg/10 g body weight for 2-week-old or dam mice. To induce Cre recombination in *Nfatc1*[CreERT2];*Rosa26*[RFP/+] mice at 2 weeks and 8 weeks of age, tamoxifen was injected for 5 consecutive days, which procedure was confirmed to induce a high recombination efficiency of *CreER*[T2] in mouse articular cartilage **Ge et al., 2016**.

## Histology and confocal microscopy imaging

Limb samples at different ages (n=3–5 mice for each age, two knee joints per animal) were fixed in 4% paraformaldehyde and processed for serial frozen sections at 8–10 µm thickness. The expression of GFP or RFP was observed using a Leica SP8 confocal microscope.

For quantifying GFP or RFP positive cells in articular cartilage, the KEYENCE BZ-X710 fluorescence microscope and software were used under the 40 x objective and cell count module. Slides were selected every 100 µm, and five slides were used for each sample (n=5 *Nfatc1*[Cre];*Rosa26*[mTmG/+] mice, three *Nfatc1*[CreERT2];*Rosa26*[RFP/+] mice). The total number of cells in each tissue was counted by combining the DAPI stain with cell morphology under phase contrast. The number of GFP$^+$ or RFP$^+$ cells was counted by combining the green or red fluorescence with DAPI. The percentage of GFP$^+$ or RFP$^+$ cells for each articular tissue was calculated by dividing the number of GFP$^+$ or RFP$^+$ cells by the total number of cells. The average of five slices from each sample was considered as the percentage of GFP$^+$ or RFP$^+$ cells in the sample.

## Cell culture and sorting

Cells were isolated from the knee of neonatal *Nfatc1*[Cre];*Rosa26*[mTmG/+] mice. Briefly, after removing the skin and most surrounding muscle tissue, the mouse knee was minced into small fragments and digested with 3 mg/ml collagenase type I (Worthington) and 4 mg/ml dispase (Roche) in complete culture media for 15 min at 37°C. After digestion, cells were passed through a 70 µm strainer and cultured for 7 days. GFP$^+$ and GFP$^-$ cells were sorted using a FACSAria fusion or FACSAria II cell sorter (BD Biosciences).

## Ex vivo assays of cell colony formation

For cell colony formation assay, single-cell suspensions of GFP$^+$ and GFP$^-$ cells (n=6 with cells from three mice, two replicates for each) were plated in 6-well-plates at indicated densities and cultured for 2 weeks. Cells were fixed in 10% neutral formalin for 15 min and stained in 1% crystal violet for 5 min. Clone numbers (>50 cells) were counted and scored blindly under the microscope.

## Ex vivo assays of osteogenic and adipogenic differentiation

The same number of cells were plated in 6-well-plates and cultured for 2 weeks in complete growth media for colony formation. Differentiation media were added to induce osteogenesis (αMEM supplemented with 10% FBS, 10 nM dexamethasone, 50 µg/ml L-ascorbic acid, and 10 mM β-glycerophosphate) for 4 weeks or adipogenesis (αMEM supplemented with 10% FBS, 100 nM dexamethasone, 50 µM indomethacin, and 5 ug/ml insulin) for 10 days. Calcium nodules and fat were visualized by staining with alizarin red and Oil Red O, respectively. Total cell clones with positive staining were

blindly counted under the microscope. The expression of osteogenic or adipogenic marker genes was examined by quantitative PCR.

## Ex vivo assay of chondrogenic differentiation

The cell pellet 3-dimension (3D) culture system for ex vivo chondrogenesis was as previously described *Embree et al., 2016*. Briefly, cells (1x10$^6$) were pelleted in 15 ml polypropylene tubes by centrifugation and cultured in chondrogenic media [DMEM high glucose supplemented with 2% FBS, 100 nM dexamethasone, 50 µg/ml L- ascorbic acid, 1% insulin, transferrin, selenium (ITS), 1 mM sodium pyruvate, 40 µg/ml L-proline, and 10 ng/ml TGF-β1] for 3 weeks. The pellets were fixed in 10% formalin and processed for paraffin embedding. Chondrogenesis was displayed by alcian blue staining and COL2A1 immunohistochemistry.

For micromass culture, 2x10$^6$ cells in 20 ul media were dropped in the center of a well in the 24-well-plate. After incubating for 2 hrs, complete culture media was added and cultured overnight. Chondrogenic media was added the next day to start the induction of chondrogenesis. Chondrogenic differentiation was evaluated by alcian blue staining and the expression of chondrocyte-associated genes.

## In vivo cell transplantation

GFP$^+$ or GFP$^-$ cells (2.0×10$^6$) mixed with Matrigel (200 µl, BD Biosciences) were injected into the dorsal skin of SCID beige mice (n=6 animals per group). Transplants were harvested after 4 weeks and fixed in 10% formalin for histological analyses. The formation of hypertrophic cartilage-like tissue in GFP$^+$ cells was determined by two experienced pathologists based on the hematoxylin stain and the hardness of the tissue when making sections.

## Immunohistochemistry

Paraffin-embedded sections (n=3 mice, two knee joints per animal for NFATc1 expression analysis or 6 cell pellets for COL2A1 expression analysis, 3–5 serial slides per sample) were digested with pepsin for 15 min at 37°C. Primary anti-Collagen Type II monoclonal antibody (Millipore Cat# MAB8887, RRID:AB_2260779), anti-NFATc1 antibody (Santa Cruz Biotechnology Cat# sc-7294, RRID:AB_2152503), or the isotype control mouse IgG1 (BioLegend Cat# 400102, RRID:AB_2891079) was incubated using a M.O.M. kit (Vector Laboratories). Sections were counterstained using Hematoxylin.

## Quantitative PCR

Total RNA was isolated using the QIAzol lysis reagent. RNA samples were treated with an RNase-Free DNase Set (QIAGEN), and equal amounts (1 µg) were used for reverse transcriptase reaction using random primers (AffinityScript QPCR cDNA Synthesis Kit). PCR primer sequences are listed in *Supplementary file 2*. All gene expression was normalized to housekeeping genes *Gapdh* and presented by $2^{-\Delta Ct}$ or $2^{-\Delta\Delta Ct}$ (*Schmittgen and Livak, 2008*).

## RNA preparation for transcriptome analysis

Total RNA was isolated from single-cell clones or bulk GFP$^+$ and GFP$^-$ cells (n=3 with cells from three different mice in each group) using QIAzol lysis reagent (Qiagen). DNA was removed using the RNase-Free DNase Set (QIAGEN). RNA was quantified with Nanodrop 2000. The integrity of RNA was evaluated using an Agilent Bioanalyzer 2100 (Agilent Technologies) and agarose gel electrophoresis.

## Library preparation and RNA-sequencing (RNA-seq)

One mg of total RNA from each sample was subjected to cDNA library construction using a NEBNext Ultra non-directional RNA Library Prep Kit for Illumina (New England Biolabs). Briefly, mRNA was enriched using oligo(dT) beads followed by two rounds of purification and fragmented randomly by adding the fragmentation buffer. The first-strand cDNA was synthesized using random hexamers primer, after which a custom second-strand synthesis buffer (Illumina), dNTPs, RNase H, and DNA polymerase I were added to generate the second-strand (ds cDNA). After a series of terminal repairs, polyadenylation, and sequencing adaptor ligation, the double-stranded cDNA library was completed following size selection and PCR enrichment.

The resulting 250–350 bp insert libraries were quantified using a Qubit 2.0 fluorometer (Thermo Fisher Scientific) and quantitative PCR. The size distribution was analyzed using the Agilent Bioanalyzer 2100. An equal amount of each RNA-Seq library was sequenced on an Illumina HiSeq 4000 Platform (Illumina) using a paired-end 150 run (2x150 bases).

## Bioinformatics analysis

Paired-end clean reads were aligned to mouse genome GRCm38/mm10 using STAR (v2.5) *Dobin et al., 2013*. HTSeq v0.6.1 was used to count the read numbers mapped to each gene. And then FPKM of each gene was calculated based on the length of the gene and read counts mapped to this gene *Trapnell et al., 2010*. Differential expression analysis between two groups was performed using the DESeq2 R package (2_1.6.3) *Anders and Huber, 2010*, which provides statistical routines for determining differential expression in digital gene expression data using a model based on the negative binomial distribution. The resulting P-values were adjusted using Benjamini and Hochberg's approach for controlling the False Discovery Rate (FDR). Genes with an adjusted *P*-value less than 0.05 found by DESeq2 were assigned as differentially expressed. Gene Ontology (GO) enrichment analysis of differentially expressed genes was implemented by the clusterProfiler R package *Yu et al., 2012*, in which gene length bias was corrected. GO terms with an adjusted P value less than 0.05 were considered to be significant.

## Flow cytometry

The following antibodies of cell surface molecules were used: Allophycocyanin (APC) anti-mouse CD45 (Thermo Fisher Scientific Cat# MCD4505, RRID:AB_10376146), anti-mouse CD31 (BioLegend Cat# 102410, RRID:AB_312905), anti-mouse CD166 (Thermo Fisher Scientific Cat# 17-1661-82, RRID:AB_2573170), anti-mouse CD200 (BioLegend Cat# 123810, RRID:AB_10900447), anti-mouse SCA1 (BioLegend Cat# 122511, RRID:AB_756196), anti-mouse CD105 (BioLegend Cat# 120414, RRID:AB_2277914); APC Rat IgG2a, κ isotype control antibody (BioLegend Cat# 400512, RRID:AB_2814702), APC Rat IgG2b, κ isotype control antibody (BioLegend Cat# 400612, RRID:AB_326556), APC Rat IgG1, κ isotype control (BioLegend Cat# 400412, RRID:AB_326518); Alexa Fluor 647 anti-mouse CD9 (BioLegend Cat# 124810, RRID:AB_2076037), Alexa Fluor 647 Rat IgG2a, κ isotype control antibody (BioLegend Cat# 400526, RRID:AB_2864284).

Cells (n=3 with cells from three different mice) were stained with antibodies or IgG isotype controls for 30 min at room temperature. Stained cells were analyzed on a FACSCalibur or BD LSR II flow cytometer (BD Biosciences). Positive cells were gated based on both unstained and isotype-matched IgG-stained cells. Data analysis was performed using FlowJo software (BD Biosciences).

## CRISPR/Cas9 lentivirus production

Pairs of CRISPR guide RNA oligos (*Nfatc1* single guide RNA [sgRNA] targeting TACGAGCTTCGG ATCGAGGT on exon 3) were annealed and cloned into the BsmBI sites of lentiCRISPR V2-blasticidin vector (a gift from Dr. Lizhi He at Harvard Medical School). CRISPR lentiviral plasmid and lentiviral packaging plasmids (pCMV-dR8.2 dvpr and pCMV-VSV-G; Addgene) were transfected into 293T cells. Supernatants were harvested and filtered through a 0.45 μm filter 2.5 days after transfection. GFP+ cells were infected with *Nfatc1*-CRISPR lentivirus. The lentivirus infected cells were selected using 5 μg/ml blasticidin for 5 days before further analyses.

## caNFATc1 retrovirus production

The retroviral expression vectors pMSCV-GFP and pMSCV-caNFATc1 have been previously described *Horsley et al., 2008*. Recombinant retroviruses were produced by co-transfecting either the pMSCV-GFP or pMSCV-caNFATc1 vectors together with retroviral packing plasmids gag/pol and pVSV-G (Addgene) into Phoenix cells using Effectene Transfection Reagent (QIAGEN). Supernatants were harvested and filtered through a 0.45 μm filter 48 hr after transfection. GFP+ cells were infected by adding retroviral supernatant with 4 μg/ml polybrene for 72 hr before further analyses.

## Western blotting

Western blotting was performed using 4–20% Mini-PROTEAN TGX precast protein gels with the following antibodies anti-NFATc1 (1:1000, Santa Cruz Biotechnology Cat# sc-7294, RRID:AB_2152503)

and anti-GAPDH (1:2000, Cell Signaling Technology Cat# 2118, RRID:AB_561053). Fifty μg of protein were loaded for each sample.

## Searching for NFATc1 binding sites in the Col2a1 gene

FIMO *Grant et al., 2011* was used to search the mouse genomic sequence of chr15: 97970080–98038617 (coordinates are based on the reference genome mm10), which contains the mouse *Col2a1* gene and the upstream and downstream intergenic regions, for the NFATc1 motif occurrence (JASPAR motif ID MA0624.2: https://jaspar.genereg.net/matrix/MA0624.2/) with the following command:

    fimo –oc . --verbosity 1 --thresh 1.0E-4 NFATc1_MA0624.2.meme sequences.fa.

As a result, 20 NFATc1 motifs (*P*-value <0.0001) were detected in this region. The FIMO output (.gff file) was visualized in the UCSC Genome Browser using the mouse reference genome mm10, together with the ENCODE Registry of candidate cis-Regulatory Elements (cCRE) and the regulatory elements from the ORegAnno, a community-driven resource for curated regulatory annotation (ENCODE cCRE and ORegAnno data are built-in annotations available in the UCSC Genome Browser). One of the 20 NFATc1 motif occurrences that overlap with the *Col2a1* promoter region is highlighted in *Figure 5—figure supplement 1B*.

## CUT&RUN-qPCR

Sorted GFP+ Cells (1x10$^5$ cells/reaction, n=3 with cells from three different mice) were incubated and combined with magnetic beads precoated with concanavalin A (Vazyme). Cells were permeabilized in antibody buffer containing 0.05% digitonin and incubated with Go-ChIP-Grade purified anti-NFATc1 antibody (10 μg/ml, BioLegend) or purified mouse IgG1, κ isotype ctrl antibody (10 μg/ml, BioLegend) overnight at 4°C. After washing in chilled digitonin buffer, cells were incubated with protein G fused Micrococcal Nuclease (pG-MNase, Vazyme) on ice for 1 hr. Subsequently, 100 mM CaCl2 solution was added to activate MNase and cleave chromatin. After stopping buffer was added, DNA fragments in the supernatant were extracted using a Qiagen MinElute PCR purification kit. The following primer sequences were used to amplify the fragment of the *Col2a1* gene that contains the potential NFATc1 binding site: forward, 5'-TTTGGAGCGACCGGGAGCATAT-3'; reverse, 5'-GGGTCTCTACCGCTCCC TCA TG-3'. The signals obtained from each immunoprecipitation are expressed as a percent of the total input chromatin and normalized with the Sample Normalization Primer Set for Spike-In DNA (Vazyme).

## Luciferase assay

The transcription activity of *Col2a1* was determined by co-transfecting ATDC5 cells (Sigma-Aldrich RRID:CVCL_3894), which had been infected with *Nfatc1*-CRISPR lentivirus or caNFATc1 retrovirus, with 1 g of the reporter plasmid (pGL2B-COL2-6.5E309, a gift from Dr. Mary Goldring) and 10 ng of pRL Renilla luciferase plasmid using the Effectene transfection reagent (Qiagen). Cells were transfected for 8 hrs and then cultured for 24 hrs. The activity of the Firefly luciferase was measured and normalized to Renilla luciferase activity using the Dual-Luciferase Reporter Assay System (Promega).

## Statistical analyses

All data are presented as mean ± SD. The normality and the equal variance of data sets were tested using the Shapiro-Wilk test and F test, respectively. Data were determined to be normally distributed and have equal variance unless specified otherwise. Differences between the two groups were evaluated by the two-tailed Student's *t*-test. For counting data of cell colonies, comparisons were performed using the nonparametric Mann-Whitney test. Analyses of multiple groups were performed using two-way ANOVA followed by Sidak's test for between-group comparisons. All analyses were performed using Prism 9.2.1 (GraphPad).

# Acknowledgements

We thank Dr. Antonios Aliprantis for helpful discussions and critically reading the manuscript; Dr. Ruirui Shi for collecting parts of these data; Dr. Lizhi He for generating the CRISPR/CAS9 lentivirus; Dr. Yidong Wang for assistance in obtaining the *Nfatc1*[Cre] mouse strain, Dr. Yulong He for assistance in obtaining the *Nfatc1*[CreERT2] mouse strain, and Drs. Guangchuang Yu and Haibo Liu for suggestions of statistical analyses.

# Additional information

### Funding

| Funder | Grant reference number | Author |
|---|---|---|
| National Natural Science Foundation of China | 81100767 | Xianpeng Ge |
| Beijing Natural Science Foundation | 5222008 | Xianpeng Ge |
| Natural Science Foundation of Capital Medical University | 1220010146 | Xianpeng Ge |
| Outstanding Young Researcher Award of Beijing Municipality | | Xianpeng Ge |
| Outstanding Researcher Award of Xuanwu Hospital Capital Medical University | | Xianpeng Ge |

The funders had no role in study design, data collection and interpretation, or the decision to submit the work for publication.

### Author contributions

Fan Zhang, Yuanyuan Wang, Data curation, Investigation; Ying Zhao, Resources; Manqi Wang, Investigation; Bin Zhou, Bin Zhou, Resources, Writing – review and editing; Xianpeng Ge, Conceptualization, Data curation, Formal analysis, Supervision, Funding acquisition, Investigation, Visualization, Writing – original draft, Writing – review and editing

### Author ORCIDs

Xianpeng Ge (iD) http://orcid.org/0000-0002-1291-2096

### Ethics

All animal studies followed the recommendations of the Guide for the Care and Use of Laboratory Animals of the U.S. National Institutes of Health and were approved by Institutional Animal Care and Use Committee at Capital Medical University (protocol #: AEEI-2022-036).

### Decision letter and Author response

Decision letter https://doi.org/10.7554/eLife.81569.sa1
Author response https://doi.org/10.7554/eLife.81569.sa2

---

# Additional files

### Supplementary files

- Supplementary file 1. Expression of cell surface markers in GFP⁺ and GFP⁻ cells.
- Supplementary file 2. Primer sequences for quantitative PCR.
- MDAR checklist

### Data availability

All data generated or analyzed during this study are included in the manuscript and supporting file. The raw datasets of RNA-seq are available in Dryad Digital Repository (https://doi.org/10.5061/dryad.2fqz612rw).

The following dataset was generated:

| Author(s) | Year | Dataset title | Dataset URL | Database and Identifier |
|---|---|---|---|---|
| Zhang F, Wang Y, Zhao Y, Wang M, Zhou B, Zhou B, Ge X | 2023 | Data for: NFATc1 marks articular cartilage progenitors and negatively determines articular chondrocyte differentiation | https://doi.org/10.5061/dryad.2fqz612rw | Dryad Digital Repository, 10.5061/dryad.2fqz612rw |

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
