## [Editor Report]

NFATc1 was known as a crucial regulator in osteoclast differentiation. The current study presented surprising and novel findings, showing the specific expression of NFATc1 in articular cartilage and its function in cartilage biology. This is a significant discovery since it will help us understand the regulatory mechanism of articular chondrocyte differentiation and the development of osteoarthritis disease.

---

## [Decision Letter]

**Decision letter after peer review:**

Thank you for submitting your article "NFATc1 negatively determines chondrocyte differentiation in articular cartilage progenitors" for consideration by *eLife*. Your article has been reviewed by 3 peer reviewers, and the evaluation has been overseen by a Reviewing Editor and Mone Zaidi as the Senior Editor. The reviewers have opted to remain anonymous.

Essential revisions:

1) To determine the function of NFATc1 in articular cartilage development at postnatal stage, the authors may need to cross Nfatc1-flox mice with Col2-CreER mice to specifically determine changes in articular cartilage morphogenesis at postnatal stage.

2) To prove there are NFATc1 binding sites at the Col2a1 promoter, the authors need to perform mutational analysis and ChIP assay.

3) The authors need to repeat the FACS-based in vitro and in vivo assays.

4) The authors need to revise the title of the manuscript to reflect the finding that NFATc1 expressing progenitors give rise to most of articular chondrocytes.

*Reviewer #1 (Recommendations for the authors):*

1) Although the authors claimed they have identified a fundamental function of NFATc1 in determining articular chondrocyte differentiation, it is known that NFATc1 plays multiple roles in bone and joint. It may not be specific for articular chondrocytes.

2) I suggest that authors use other reporters to verify this finding due to non-specific background staining of green fluorescence protein (GFP)

3) To determine the function of NFATc1 in articular cartilage development at postnatal stage, the authors may consider crossing Nfatc1flox/flox mice with Col2-CreER mice to specifically determine changes in articular cartilage morphogenesis at postnatal stage.

4) Figure 5E: It seems that there are dramatic changes in growth plate cartilage in Nfatc1 KO mice. These data contradict with the authors' claim that NFATc1 affacts articular cartilage differentiation, but has no effect on growth plate cartilage.

5) Figure 5H: Mislabel. To prove there are NFATc1 binding sites at the Col2a1 promoter, the authors need to perform mutational analysis and ChIP assay.

6) Figure S1D, right panel: As described in figure legend, this picture showed Nfatc1-CreERT2;Rosa26-mTmG fl/+ mice without tamoxifen induction. It is not clear why there are so many RFP+ cells in articular cartilage area if this is a negative control. Are there significant leaking problems in Nfatc1-CreERT2 mice?

7) Figure S2: In the title of the figure legend the authors stated "Colony formation and multipotent differentiation of Nfatc1-expressing progenitors". However, I don't see any multipotent differentiation of Nfatc1-expressing progenitors in this figure.

*Reviewer #2 (Recommendations for the authors):*

1. One major concern with severe technical issues is an animal model option for FACS-based studies. FACS utilized here is predominantly based on NFATc1 Cre-mTmG rodents whereas Figure 1A and 1B displayed almost all articular cartilage cells express, or expressed, NFATc1 during developmental stage. Thus, the conclusions addressed based on this technique represented the biological behavior of all articular chondrocytes though this point was partially discussed in limitations. Should it be considered that an inducible-Cre model, such as NFATc1-CreER individuals, be taken advantage of in future FACS-based in vitro and in vivo assays to avoid these limitations. Otherwise, all these experiments downstream of FACS were just comparing chondrocytes with muscle cells, according to gene ontology results.

2. A likely nuisance of this data with previous studies is that in Figure 4, GFP- cells includes a similar proportion of CD200+ cells and a higher amount of CD105- cells compared with GFP+ cells but, conversely, shown a reduced ability in differentiation ability albeit other exclusive markers are not compared through flow cytometry in this figure, and this should be worried more considering the total loss of physiological function according to Figure S2B. Although transcriptome study uncovered these cells are mesenchymal-derived progenitors, this requires to be clarified as it clearly disobeyed the phenomenon observed in murine models according to previous documents. Further in vitro or in vivo assays are required to evaluate the conclusion of GFP+ cells were generated from their intrinsic progenitor functions or from a baseline issue consequent to loss-of-function of GFP- cells.

3. At the same time, since osteoclasts also highly express NFATc1, would it be possible to analyze some major clastogenic gene to prove how precise the dissections performed for FACS were and exclude potential osteoclast contamination?

4. The approaches of in vivo transplantation here somehow exhibited leakage of cells from the calcified organoid, pointed out by black arrows, formed as HE staining displaying some histologically identified chondrocytes at both right and lower sides. Saf-O staining also confirmed this concern by showing escaped chondrocytes outside of the Matrigel organoid. This concern is also linked to major point 1 since GFP+ cells implanted are almost all chondrocytes, it is hard to make a conclusion that NFATc1+ cells are able to form cartilage organoid in vivo. Experimental details and more appreciable results are required to be specifically discussed for the clarification of those confusions.

5. Related to point 4, one question needs to be answered: why kidney capsule transplantation was not performed in this case? Apparently, renal capsule provides a more stable environment for cellular graft compared to subcutaneous space considering the nutrient supply and the immune attack might cause unwanted outcomes.

6. According to Murphy MP., et al., Nat Med., 2020, SSCs and BCSPs that contribute to articular cartilage regeneration were defined by cells expressing or lacking a group of surface marker. Whereas in this manuscript those panels are not the same, especially ItgaV which were proven to be an important marker for cells that have capacity of self-renew and multipotent differentiation abilities. It might be worthwhile for authors to go over those panels in their previous experiments to confirm if NFATc1+ cells overlap with SSCs and/or BCSPs that could functionally confirm NFATc1+ cells as a crucial source of chondrogenic progenitors.

7. It is not appropriate to address GFP+ cells formed mineralized tissues around them simply through HE staining according to Figure 3F. A Von-Kossa staining is required for the sake of basic validation of calcified area. Also, Saf-O staining looks weird and requires to be duplicated for better images.

8. The authors reported that NFATc1-expressing progenitors generate articular chondrocytes, meniscus cells, and synovial lining cells. Do these progenitors also generate articular ligaments?

9. Does the deletion of Nfatc1 in Prrx1-cre; Nfatc1flox mice affect the ligament? It looks like the articular ligament is abnormal in Figure 5E, which should be mentioned in the manuscript.

*Reviewer #3 (Recommendations for the authors):*

1) The title does not reflect the finding that NFATc1 expressing progenitors give rise to most of articular cartilage chondrocytes. The title should be revised to reflect such finding.

2) There is a discordance between the NFATc1 expression and NFATc1 expressing cells on articular cartilage according to Figure 1 and 2. In Figure 1E, NFATc1 immunostaining of articular suggests abundance of NFATc1 expressing cells at P0. However, in Figure 2, only a limited number of cells on the superficial layer is GFP positive despite the cre-induction occurring at P0 and P1. The expected results would be GFP-positive population abundance would be reflective of NFATc1 immunostained cell distribution, which is not according to Figure 1 and 2.

3) Based on the expression pattern of NFATc1 expressing progenitors at E13.5, P0, 2 weeks, and 8 weeks (Figure 1), one wonders whether NFATc1 expressing cells at E13.5 are a subset of GDF5 expressing cells or a distinct population during that time or earlier. Articular chondrocyte progenitors are likely heterogenous and may need multiple markers to identify them. Immunofluorescence for GDF5 and whether that co-localized with NFATc1 expressing cells could potentially reveal whether these cell populations overlap.

4) Authors could further strengthen their finding that NFATc1 expressing progenitors being multipotent (Figure 3) by performing more analyses of osteoblasts and adipocytes. They already have evidence in Figure 1(A) that shows abundance of GFP-positive bone lining cells. In addition, tibial metaphyseal bone marrow adipocytes may also be positive for GFP, which cannot be ascertained from the provided data. Authors can perform immunofluorescence of FISH to identify whether these GFP-positive cells express markers of osteoblasts or adipocytes.

5) Although not in articular cartilage, NFATc1 ChIP-seq has been previously performed (Keyes et al. 2013) with the public dataset already available. Through these studies authors may be able to further narrow down the binding sites of NFATc1.

---

## [Author Response]

Essential revisions:1) To determine the function of NFATc1 in articular cartilage development at postnatal stage, the authors may need to cross Nfatc1-flox mice with Col2-CreER mice to specifically determine changes in articular cartilage morphogenesis at postnatal stage.

The Col2-Cre;Nfatc1-flox (Nfatc1Col2) mouse model has been reported in a previous study1. These animals do not develop gross and histological defects in the articular cartilage. Compared to wild-type controls, they also do not show differences in osteoarthritis progression by destabilization of the medial meniscus. These results indicate that NFATc1 is dispensable in Col2-expressing articular chondrocytes. In fact, these results match our findings in the current study that NFATc1 mainly characterizes articular cartilage progenitors, and its expression decreases with articular chondrocyte differentiation. As Nfatc1 is deleted in all Col2-expressing cells, including postnatal articular chondrocytes, in the Nfatc1Col2 mouse model, we think it is not necessary to further analyze the tamoxifen-induced Col2-CreER;Nfatc1-flox mouse model.

In reference to the reviewer’s comment, we have integrated the previous results from Nfatc1Col2 mice into the manuscript (Lines 253-258 in the merged PDF of the revised manuscript). Specifically, we state:

“Note that a previous study showed that Nfatc1 deletion in Col2-expressing cells (Nfatc1Col2 mice) does not affect articular cartilage integration or osteoarthritis progression induced by destabilization of the medial meniscus1, suggesting that NFATc1 is dispensable in Col2-expressing differentiated articular chondrocytes. Therefore, NFATc1 may primarily function in articular cartilage progenitors to regulate articular chondrocyte differentiation.”

2)To prove there are NFATc1 binding sites at the Col2a1 promoter, the authors need to perform mutational analysis and ChIP assay.

To identify the specific binding site of NFATc1 in the regulatory region of the *Col2a1* gene, we additionally performed a FIMO analysis^2^, which localizes an NFATc1 motif in chr15: 98004609-98004620 that overlaps with the *Col2a1* promoter region (Figure 5—figure supplement 1B). Furthermore, we use the CUT&RUN-qPCR technique, an alternative method of ChIP-qPCR, to confirm that this site is indeed bound by NFATc1 (Figure 5H).

These new results have been added to the manuscript (Lines 265-270 in the merged PDF of the revised manuscript). Specifically, we add the following sentences:

“To narrow down the NFATc1 binding sites in the regulatory region of the *Col2a1* gene, we performed a FIMO analysis and identified an NFATc1 motif located in mouse chr15: 98004609-98004620 that overlaps with the *Col2a1* promoter region (Figure 5—figure supplement 1B). Furthermore, we verified the binding of NFATc1 to this site using the technique of cleavage under targets and release using nuclease (CUT&RUN) (Figure 5H).”

3)The authors need to repeat the FACS-based in vitro and in vivo assays.

This comment has been addressed in the following responses to specific comments of reviewer #2.

4)The authors need to revise the title of the manuscript to reflect the finding that NFATc1 expressing progenitors give rise to most of articular chondrocytes.

In reference to the reviewer’s suggestion, we have changed the title of the manuscript to: “NFATc1 marks articular cartilage progenitors and negatively determines articular chondrocyte differentiation”

Reviewer #1 (Recommendations for the authors):1) Although the authors claimed they have identified a fundamental function of NFATc1 in determining articular chondrocyte differentiation, it is known that NFATc1 plays multiple roles in bone and joint. It may not be specific for articular chondrocytes.

We agree with the reviewer that the function of NFATc1 is not specific for articular chondrocytes. However, the finding that NFATc1 determines the differentiation of articular chondrocytes during embryonic development and postnatal growth is highly novel, which is more fundamental compared to our previous finding that deletion of NFATc1 in entheseal progenitors causes osteochondroma formation^3^. Note that the functions of NFATc1 in osteoclasts and osteoblasts have been mentioned in the introduction of the manuscript (Lines 83-85 in the merged PDF of the revised manuscript).

2) I suggest that authors use other reporters to verify this finding due to non-specific background staining of green fluorescence protein (GFP)

All GFP fluorescence in this study was observed directly after tissue fixation and sectioning from the genetic mouse model *Nfatc1-Cre;Rosa26-mTmG^fl/+^* or tamoxifen-induced *Nfatc1-CreER^T2^;Rosa26-mTmG^fl/+^* mice. No additional immunostaining was performed for GFP. Importantly, Cre negative *Rosa26-mTmG^fl/+^* control mice or *Nfatc1-CreER^T2^;Rosa26-mTmG^fl/+^*mice without tamoxifen pulse were carefully examined and no GFP fluorescence was observed in these animals (Figure 1—figure supplement 1D), indicating that the GFP fluorescence observed in experimental animals is specific.

3) To determine the function of NFATc1 in articular cartilage development at postnatal stage, the authors may consider crossing Nfatc1flox/flox mice with Col2-CreER mice to specifically determine changes in articular cartilage morphogenesis at postnatal stage.

This comment has been addressed in the above response to point #1 of the Essential Revisions.

4) Figure 5E: It seems that there are dramatic changes in growth plate cartilage in Nfatc1 KO mice. These data contradict with the authors' claim that NFATc1 affacts articular cartilage differentiation, but has no effect on growth plate cartilage.

The thickness of the growth plate presented in the image is greatly affected by the layer and angle of the tissue sections. We review again all serial sections from *Prrx1-Cre;Nfatc1^fl/fl^* and *Prrx1-Cre;Nfatc1^fl/+^* mice. There is no obvious difference in terms of growth plate cartilage (see Author response image 1).

**Author response image 1. sa2fig1:** 

5) Figure 5H: Mislabel. To prove there are NFATc1 binding sites at the Col2a1 promoter, the authors need to perform mutational analysis and ChIP assay.

We performed the FIMO analysis and the CUT&RUN-qPCR experiment, which identify a specific binding site of NFATc1 in the *Col2a1* promoter region. We refer the editors and the reviewer to our above response to point #2 of the Essential Revisions in which we address this comment in more detail.

We have corrected the label for Figure 5H (now it is Figure 5I) in the figure legend.

6) Figure S1D, right panel: As described in figure legend, this picture showed Nfatc1-CreERT2;Rosa26-mTmG fl/+ mice without tamoxifen induction. It is not clear why there are so many RFP+ cells in articular cartilage area if this is a negative control. Are there significant leaking problems in Nfatc1-CreERT2 mice?

In *Rosa26-mTmG* reporter mice, cells constitutively express tdTomato without Cre recombination, while cells in this reporter mouse will express GFP fluorescence with Cre recombination. Therefore, it is rational that red fluorescence (tdTomato) was observed in *Nfatc1-CreER^T2^;Rosa26-mTmG^fl/+^* mice without the pulse of tamoxifen.

7) Figure S2: In the title of the figure legend the authors stated "Colony formation and multipotent differentiation of Nfatc1-expressing progenitors". However, I don't see any multipotent differentiation of Nfatc1-expressing progenitors in this figure.

Sorry for the confusion. We have changed the title of Figure S2 (now it is Figure 3—figure supplement 1) as follows: “Colony formation assay and in vivo transplantation of GFP^-^ cells”.

Reviewer #2 (Recommendations for the authors):1. One major concern with severe technical issues is an animal model option for FACS-based studies. FACS utilized here is predominantly based on NFATc1 Cre-mTmG rodents whereas Figure 1A and 1B displayed almost all articular cartilage cells express, or expressed, NFATc1 during developmental stage. Thus, the conclusions addressed based on this technique represented the biological behavior of all articular chondrocytes though this point was partially discussed in limitations. Should it be considered that an inducible-Cre model, such as NFATc1-CreER individuals, be taken advantage of in future FACS-based in vitro and in vivo assays to avoid these limitations. Otherwise, all these experiments downstream of FACS were just comparing chondrocytes with muscle cells, according to gene ontology results.

We thank the reviewer for this suggestion. Cells for FACS analysis in this study were from neonatal *Nfatc1- Cre;Rosa26-mTmGfl/+* mice (postnatal day 0), at which age the incipient articular cartilage consists mainly of progenitor cells that have not yet developed the phenotypic and molecular characteristics of differentiated articular chondrocytes^4^. The progenitor cell properties of these cells are well demonstrated in vivo and ex vivo (Figures 2 and 3). Furthermore, the FACS analysis in Figure 4E and F aimed primarily to verify the expression of several surface molecular markers based on the RNA-seq results in NFATc1^+^ articular cartilage progenitors.

The heterogeneity of the GFP^+^ cell population is mainly caused by different cell lineages in the joint (articular cartilage, meniscus, synovium, and ligament) and different stages of progenitor cell development, which will probably still exist in the *Nfatc1-CreER^T2^* mouse model after multiple pulses of tamoxifen. As discussed in the manuscript (Lines 368-373 in the merged PDF of the revised manuscript), the heterogeneity will not affect the conclusion of the current manuscript. Future studies combining single-cell analysis and the tamoxifen-induced *Nfatc1-CreER^T2^* mouse model would be able to dissect the heterogeneity of the progenitor cell population.

2. A likely nuisance of this data with previous studies is that in Figure 4, GFP- cells includes a similar proportion of CD200+ cells and a higher amount of CD105- cells compared with GFP+ cells but, conversely, shown a reduced ability in differentiation ability albeit other exclusive markers are not compared through flow cytometry in this figure, and this should be worried more considering the total loss of physiological function according to Figure S2B. Although transcriptome study uncovered these cells are mesenchymal-derived progenitors, this requires to be clarified as it clearly disobeyed the phenomenon observed in murine models according to previous documents. Further in vitro or in vivo assays are required to evaluate the conclusion of GFP+ cells were generated from their intrinsic progenitor functions or from a baseline issue consequent to loss-of-function of GFP- cells.

The origin of the GFP^+^ articular cartilage progenitors mentioned by the reviewer in this comment is an interesting question that needs to be further addressed in future studies. Studies have shown that there could be multiple different pools of stem cells in the generation and maintenance of the musculoskeletal system^5-8^. Since the NFATc1^+^ articular cartilage progenitors we identified in this study do not generate the mouse growth plate primordium during embryonic development (Figure 1A and Figure 1—figure supplement 1A), they probably represent a distinct lineage of skeletal stem cells from those mentioned by the reviewer in previous documents. Consequently, it remains unclear whether these different lineages of skeletal stem cells share the same profile of surface molecules. Although our transcriptome data and FACS results in Figure 4D, E and F provide certain insights into the molecular characteristics of the GFP^+^ and GFP^-^ progenitor cells, further studies need to dissect the developmental hierarchy and the surface molecular profile of NFATc1^+^ articular cartilage progenitors and systemically compare them with other populations of skeletal stem cells.

In reference to the reviewer’s comment, we add the following sentence to the Discussion section of the manuscript (Lines 388-390 in the merged PDF of the revised manuscript):

“Lastly, the origin of NFATc1^+^ articular cartilage progenitors should be further explored, which will be critical to better understand the basic mechanism of articular cartilage development and leverage it for articular cartilage regeneration.”

3. At the same time, since osteoclasts also highly express NFATc1, would it be possible to analyze some major clastogenic gene to prove how precise the dissections performed for FACS were and exclude potential osteoclast contamination?

Osteoclasts are considered cells of hematopoietic origin^9^. Both RNA-seq data (Supplementary file 1) and FACS analysis (Figure 4—figure supplement 2A) showed negative staining for CD11B and CD45 in GFP^+^ and GFP^-^ cells, which can exclude osteoclast contamination in both cell populations.

4. The approaches of in vivo transplantation here somehow exhibited leakage of cells from the calcified organoid, pointed out by black arrows, formed as HE staining displaying some histologically identified chondrocytes at both right and lower sides. Saf-O staining also confirmed this concern by showing escaped chondrocytes outside of the Matrigel organoid. This concern is also linked to major point 1 since GFP+ cells implanted are almost all chondrocytes, it is hard to make a conclusion that NFATc1+ cells are able to form cartilage organoid in vivo. Experimental details and more appreciable results are required to be specifically discussed for the clarification of those confusions.

Sorry for the confusion. Note that the GFP^+^ cell population enriched with articular cartilage progenitors could be at different stages of development. The development of postnatal articular cartilage is highly dependent on the increased volume of articular chondrocytes^10^. The colony formation capacity of postnatal articular cartilage progenitors is limited to 3-6 cells/clone (Figure 2A and reference^10^). We refer the editors and the reviewer to our response to comment #1 of this reviewer, in which we explain the population of GFP^+^ cells and the characteristics of articular cartilage development in more detail.

5. Related to point 4, one question needs to be answered: why kidney capsule transplantation was not performed in this case? Apparently, renal capsule provides a more stable environment for cellular graft compared to subcutaneous space considering the nutrient supply and the immune attack might cause unwanted outcomes.

We thank the reviewer for this suggestion. It should be noted that articular cartilage is a blood vessel and nerve- free tissue, and the development of articular cartilage occurs right underneath the skin. Therefore, we consider that the subcutaneous space is closer to the native environment of articular cartilage development. The Matrigel can provide the structural organization of the extracellular matrix and essential growth factors for chondrogenesis.

Kidney capsule transplantation is a very useful model in the analysis of osteogenesis and chondrogenesis of skeletal stem cells from the bone marrow or growth plate. It would be interesting to try this method for NFATc1^+^ skeletal stem cells and compare it with subcutaneous transplantation in future studies.

6. According to Murphy MP., et al., Nat Med., 2020, SSCs and BCSPs that contribute to articular cartilage regeneration were defined by cells expressing or lacking a group of surface marker. Whereas in this manuscript those panels are not the same, especially ItgaV which were proven to be an important marker for cells that have capacity of self-renew and multipotent differentiation abilities. It might be worthwhile for authors to go over those panels in their previous experiments to confirm if NFATc1+ cells overlap with SSCs and/or BCSPs that could functionally confirm NFATc1+ cells as a crucial source of chondrogenic progenitors.

We thank the reviewer for this suggestion. The RNA-seq data showed that both GFP^+^ and GFP^-^ cells express a high level of *Itgav* (*Cd51*). We have added the data in Supplementary file 1 and the manuscript (Line 213 in the merged PDF).

In the paper by Murphy MP, et al., cartilage formation was induced from Itgav^+^CD200^+^CD105^-^ SSCs by adding BMP2 and FGF inhibitors^11^. For this purpose, the authors induced microfracture lesions in the articular cartilage by drilling the surface of the articular cartilage down to the marrow space. In this context, the population of stem cells that regenerate cartilage remains undefined. As mentioned above in our response to this reviewer’s comment #2, the articular cartilage progenitors are probably different from the Itgav^+^CD200^+^CD105^-^ SSCs that were originally identified from the postnatal growth plate^5^. We agree with the reviewer that future studies should carefully compare molecular profiles and cartilage formation capacity between NFATc1^+^ articular cartilage progenitors and Itgav^+^CD200^+^CD105^-^ SSCs.

7. It is not appropriate to address GFP+ cells formed mineralized tissues around them simply through HE staining according to Figure 3F. A Von-Kossa staining is required for the sake of basic validation of calcified area. Also, Saf-O staining looks weird and requires to be duplicated for better images.

The formation of calcified cartilage around GFP^+^ cells was first noticed by the hardness of these tissues when making sections, and in particular, the HE staining was evaluated by two experienced pathologists (Lines 474- 476 in the merged PDF of the revised manuscript). Note that a specific method for staining calcified articular cartilage has not been well established yet. Von-Kossa staining is usually used to recognize mineralized bone matrix, which has not been widely used for staining calcified articular cartilage.

It should be noted that the Matrigel background may make the safranin-O staining different from that of native articular cartilage. The safranin-O staining in this study was repeated multiple times using sections from different animals (*n* = 6 per group).

8. The authors reported that NFATc1-expressing progenitors generate articular chondrocytes, meniscus cells, and synovial lining cells. Do these progenitors also generate articular ligaments?

Yes, NFATc1-expressing progenitors also generate cells in the articular ligaments, which has been mentioned in our previous study^3^. In reference to the reviewer’s comment, we have also highlighted the finding in Figure 2 and the manuscript.

9. Does the deletion of Nfatc1 in Prrx1-cre; Nfatc1flox mice affect the ligament? It looks like the articular ligament is abnormal in Figure 5E, which should be mentioned in the manuscript.

Yes, deletion of *Nfatc1* in *Prrx1*-Cre;*Nfatc1*-flox mice also caused abnormal ligament development and ectopic cartilage formation in articular ligaments. These findings were mentioned in our previous article^3^. In this study, we focus on a novel physiological function of NFATc1 in identifying progenitors of articular cartilage and determining articular chondrocyte differentiation.

Reviewer #3 (Recommendations for the authors):1) The title does not reflect the finding that NFATc1 expressing progenitors give rise to most of articular cartilage chondrocytes. The title should be revised to reflect such finding.

We thank the reviewer for this important suggestion. The title of the manuscript has been changed as follows: “NFATc1 marks articular cartilage progenitors and negatively determines articular chondrocyte differentiation”

2) There is a discordance between the NFATc1 expression and NFATc1 expressing cells on articular cartilage according to Figure 1 and 2. In Figure 1E, NFATc1 immunostaining of articular suggests abundance of NFATc1 expressing cells at P0. However, in Figure 2, only a limited number of cells on the superficial layer is GFP positive despite the cre-induction occurring at P0 and P1. The expected results would be GFP-positive population abundance would be reflective of NFATc1 immunostained cell distribution, which is not according to Figure 1 and 2.

The discordance in NFATc1 expression between Figures 1 and 2 could be due to the recombination efficiency of the CreER^T2^-LoxP system in Figure 2. The recombination efficiency of the CreER^T2^-LoxP system is affected by the dosage, times, and routes of administration of tamoxifen. The experiment in Figure 2 aimed to observe in vivo colony formation of NFATc1-expressing progenitors and two doses of tamoxifen in the dams were enough to label some NFATc1-expressing articular cartilage progenitors in the pups and trace their fate in vivo. Note that this dose of tamoxifen administered to dam mice may not be enough to induce full recombination of the CreER^T2^- LoxP system in pup mice.

3) Based on the expression pattern of NFATc1 expressing progenitors at E13.5, P0, 2 weeks, and 8 weeks (Figure 1), one wonders whether NFATc1 expressing cells at E13.5 are a subset of GDF5 expressing cells or a distinct population during that time or earlier. Articular chondrocyte progenitors are likely heterogenous and may need multiple markers to identify them. Immunofluorescence for GDF5 and whether that co-localized with NFATc1 expressing cells could potentially reveal whether these cell populations overlap.

We thank the reviewer for this nice suggestion. We tried immunofluorescence staining for GDF5 in E13.5 mouse embryos. Unfortunately, we were unable to locate an antibody for specific staining of GDF5 in mouse tissue. However, by combining our results in this study with previous lineage tracing data in *Gdf5*-Cre and *Gdf5*-CreER reporter mice, which showed that most articular chondrocytes are derived from GDF5-expressing articular cartilage progenitors, NFATc1-expressing articular cartilage progenitors are probably a subset of GDF5- expressing progenitors. we are continuing to localize a specific antibody for GDF5 staining and will address this question in more detail in the next study.

According to this comment of the reviewer, we have discussed this point in the manuscript (Lines 307-310 in the merged PDF). Specifically, we state: “Furthermore, based on the distribution of GDF5^+^ progenitors during synovial joint development and their contribution to the formation of most articular chondrocytes in previous reports^10, 12^, NFATc1-expressing articular cartilage progenitors are probably a subset of GDF5^+^ progenitors.”

We agree with the reviewer that multiple markers, especially surface molecules, are necessary to identify articular cartilage progenitors. An ongoing study in our group is dissecting the developmental hierarchy of NFATc1-expressing articular cartilage progenitors that will be able to reveal specific markers, especially surface markers, to identify articular cartilage progenitors.

4) Authors could further strengthen their finding that NFATc1 expressing progenitors being multipotent (Figure 3) by performing more analyses of osteoblasts and adipocytes. They already have evidence in Figure 1(A) that shows abundance of GFP-positive bone lining cells. In addition, tibial metaphyseal bone marrow adipocytes may also be positive for GFP, which cannot be ascertained from the provided data. Authors can perform immunofluorescence of FISH to identify whether these GFP-positive cells express markers of osteoblasts or adipocytes.

We thank the reviewer for the insightful suggestion. The focus of the current study is to report the findings that NFATc1 identifies articular cartilage progenitors and the function of NFATc1 in determining articular chondrocyte differentiation. The contribution of NFATc1^+^ skeletal stem cells to osteoblasts and bone marrow adipocytes is beyond the scope of the current study. However, the authors recognize that it will be critical to address these questions in future studies.

5) Although not in articular cartilage, NFATc1 ChIP-seq has been previously performed (Keyes et al. 2013) with the public dataset already available. Through these studies authors may be able to further narrow down the binding sites of NFATc1.

We thank the reviewer for this suggestion. To narrow down the binding site of NFATc1 in the *Col2a1* gene promoter, we instead performed a FIMO analysis^2^ and identified an NFATc1 motif that is located in mouse chr15: 98004609-98004620 and overlaps with the *Col2a1* promoter region (Figure 5—figure supplement 1B). Furthermore, we validated the binding of NFATc1 to this site by the technique of cleavage under targets and release using nuclease (CUT&RUN) (Figure 5H). These results have been added to the manuscript (Lines 265-270 in the merged PDF of the revised manuscript).

References

1. Greenblatt MB, Ritter SY, Wright J, Tsang K, Hu D, Glimcher LH, et al. NFATc1 and NFATc2 repress spontaneous osteoarthritis. Proc Natl Acad Sci U S A 2013; 110: 19914-19919.

2. Grant CE, Bailey TL, Noble WS. FIMO: scanning for occurrences of a given motif. Bioinformatics 2011; 27: 1017-1018.

3. Ge X, Tsang K, He L, Garcia RA, Ermann J, Mizoguchi F, et al. NFAT restricts osteochondroma formation from entheseal progenitors. JCI Insight 2016; 1: e86254.

4. Decker RS, Koyama E, Pacifici M. Articular Cartilage: Structural and Developmental Intricacies and Questions. Curr Osteoporos Rep 2015; 13: 407-414.

5. Chan CK, Seo EY, Chen JY, Lo D, McArdle A, Sinha R, et al. Identification and specification of the mouse skeletal stem cell. Cell 2015; 160: 285-298.

6. Debnath S, Yallowitz AR, McCormick J, Lalani S, Zhang T, Xu R, et al. Discovery of a periosteal stem cell mediating intramembranous bone formation. Nature 2018; 562: 133-139.

7. Mizuhashi K, Ono W, Matsushita Y, Sakagami N, Takahashi A, Saunders TL, et al. Resting zone of the growth plate houses a unique class of skeletal stem cells. Nature 2018; 563: 254-258.

8. Koyama E, Shibukawa Y, Nagayama M, Sugito H, Young B, Yuasa T, et al. A distinct cohort of progenitor cells participates in synovial joint and articular cartilage formation during mouse limb skeletogenesis. Dev Biol 2008; 316: 62-73.

9. Bar-Shavit Z. The osteoclast: a multinucleated, hematopoietic-origin, bone-resorbing osteoimmune cell. J Cell Biochem 2007; 102: 1130-1139.

10. Decker RS, μm HB, Dyment NA, Cottingham N, Usami Y, Enomoto-Iwamoto M, et al. Cell origin, volume and arrangement are drivers of articular cartilage formation, morphogenesis and response to injury in mouse limbs. Dev Biol 2017; 426: 56-68.

11. Murphy MP, Koepke LS, Lopez MT, Tong X, Ambrosi TH, Gulati GS, et al. Articular cartilage regeneration by activated skeletal stem cells. Nat Med 2020; 26: 1583-1592.

Decker RS. Articular cartilage and joint development from embryogenesis to adulthood. Semin Cell Dev Biol 2017; 62: 50-56.